# Vacancy-mediated anomalous phononic and electronic transport in defective half-Heusler ZrNiBi

Wuyang Ren[1,2,6], Wenhua Xue[3,6], Shuping Guo [4,6], Ran He[4], Liangzi Deng [2], Shaowei Song[2], Andrei Sotnikov[4], Kornelius Nielsch [4], Jeroen van den Brink [4], Guanhui Gao[5], Shuo Chen [2], Yimo Han [5], Jiang Wu [1], Ching-Wu Chu [2], Zhiming Wang [1]✉, Yumei Wang[3]✉ & Zhifeng Ren [2]✉

Studies of vacancy-mediated anomalous transport properties have flourished in diverse fields since these properties endow solid materials with fascinating photoelectric, ferroelectric, and spin-electric behaviors. Although phononic and electronic transport underpin the physical origin of thermoelectrics, vacancy has only played a stereotyped role as a scattering center. Here we reveal the multifunctionality of vacancy in tailoring the transport properties of an emerging thermoelectric material, defective n-type ZrNiBi. The phonon kinetic process is mediated in both propagating velocity and relaxation time: vacancy-induced local soft bonds lower the phonon velocity while acoustic-optical phonon coupling, anisotropic vibrations, and point-defect scattering induced by vacancy shorten the relaxation time. Consequently, defective ZrNiBi exhibits the lowest lattice thermal conductivity among the half-Heusler family. In addition, a vacancy-induced flat band features prominently in its electronic band structure, which is not only desirable for electron-sufficient thermoelectric materials but also interesting for driving other novel physical phenomena. Finally, better thermoelectric performance is established in a ZrNiBi-based compound. Our findings not only demonstrate a promising thermoelectric material but also promote the fascinating vacancy-mediated anomalous transport properties for multidisciplinary explorations.

Transport properties of a solid material can be mediated by structural defects that breed multifarious novel behaviors for applications in thermotics, electronics, optics, magnetics, etc[1-4]. Vacancy has been identified as an important type of defect and its versatility has been manifested in diverse areas, e.g., anomalous $d^0$ ferromagnetism in dielectric oxides[5,6], the delicate quantum nature of color centers in diamond[7,8], and Rashba spin-orbit coupling in inversion-symmetry-breaking systems[9]. However, the functionality of vacancy seems sterile in thermoelectrics, where it basically acts as a phonon scattering center[10,11]. Recent investigations of vacancy-ordering-induced topological electronic transition in $Eu_2ZnSb_2$[12] and vacancy-related strain engineering in PbTe[13] are particularly impressive for their high thermoelectric performance results, signifying that vacancy could play a

[1]Institute of Fundamental and Frontier Sciences, University of Electronic Science and Technology of China, Chengdu 610054, People's Republic of China. [2]Department of Physics and Texas Center for Superconductivity at the University of Houston (TcSUH), Houston, TX 77204, USA. [3]Beijing National Laboratory for Condensed Matter Physics, Institute of Physics, Chinese Academy of Science, Beijing 100190, People's Republic of China. [4]Leibniz Institute for Solid State and Materials Research, Dresden 01069, Germany. [5]Department of Materials Science and Nano-Engineering, Rice University, Houston, TX 77005, USA. [6]These authors contributed equally: Wuyang Ren, Wenhua Xue, Shuping Guo. ✉e-mail: zhmwang@uestc.edu.cn; wangym@iphy.ac.cn; zren@uh.edu

more pivotal role in advancing thermoelectrics beyond its phonon scattering functionality.

The emerging 19-electron half-Heusler (HH) materials provide an ideal platform for exploration of vacancy-mediated anomalous transport properties since their inherent vacancy is vital to their structural stability[14,15]. In contrast to a typical 18-electron HH with unfilled anti-bonding states, a stoichiometric 19-electron HH is electron-sufficient and its densely occupied anti-bonding states will thus increase its instability. Following the valence-balanced rule, the defective structure of a stoichiometric 19-electron HH accommodating a large concentration of cationic vacancies is energetically preferred, and a number of stable compounds have also been experimentally reported[16–18]. A thorough examination of the thermodynamic stability and delicate control of vacancy concentration has been undertaken to discover new defective 19-electron HHs that extend the family of Heuslers[19–21]. Meanwhile, the major drawback of thermoelectric HHs—their intrinsically high lattice thermal conductivity ($\kappa_{lat}$)—has been addressed to some extent (Fig. 1)[18,22,23]. For instance, the room-temperature $\kappa_{lat}$ of defective $Nb_{0.8}CoSb$ is ~4.2 W m$^{-1}$ K$^{-1}$, while that of NbCoSn is ~7.7 W m$^{-1}$ K$^{-1}$. There is no doubt that vacancy-mediated short phonon relaxation time will give rise to such reduced $\kappa_{lat}$. Nonetheless, the question whether vacancy endows defective 19-electron HHs with other anomalous transport properties remains nearly unaddressed. It is thus compelling to dig more deeply into the multifunctionality of vacancy in tailoring the phononic and electronic transport properties of defective 19-electron HHs.

Here we first describe an emerging defective ZrNiBi with a pure HH phase. Its randomly distributed zirconium vacancy is revealed using high-angle annular dark-field scanning transmission electron microscopy. Another distinctive structural characteristic is the existence of twofold twinning, which is rarely reported in HHs. More intriguingly, this material is distinguished among HHs due to its anomalous phononic and electronic transport mediated by its vacancy. An exceptionally low $\kappa_{lat}$ of ~1.4 W m$^{-1}$ K$^{-1}$ at room temperature is obtained in defective ZrNiBi, the lowest among the HH family and comparable to that of other outstanding thermoelectric materials (Fig. 1). Besides the contribution of vacancy-mediated short phonon relaxation time, the phononic dynamics elucidates that the origin of such low $\kappa_{lat}$ also correlates with vacancy-induced local soft bonds, low-lying optical phonons, and anisotropic lattice vibrations. In addition, the electronic band structure is modified by the cationic vacancy so that a flat band is observed near the conductive band minimum, which is unique among HHs and of great importance in physics. The flat band results in a large density of states near the band edge, which is particularly desirable for the electron-sufficient thermoelectric materials since large density of states effective mass requires high optimal carrier concentration. Finally, further carrier concentration optimization endows the defective ZrNiBi-based material with better thermoelectric properties, allowing it to outperform various typical n-type HHs. We believe that our findings of vacancy-mediated anomalous transport in defective ZrNiBi will stimulate the discovery of novel phenomena and the development of both innovative theories and potential technological applications.

## Results

### Crystallographic features

According to the Open Quantum Materials Database[24], the stoichiometric ZrNiBi with a valence electron count (VEC) of 19 lies above the convex hull (hull distance is ~0.065 eV/atom) constructed by connecting the vertices based on the $T = 0$ K formation energies of all phases in its composition space, indicating that it is thermodynamically unstable. This is consistent with the X-ray diffraction (XRD) results shown in Fig. 2a, in which additional ZrNi alloy peaks can be observed. On the other hand, although $Zr_{0.75}NiBi$ is valence balanced [net valence, NV = 3 (0.75Zr$^{4+}$ s$^0$d$^0$) − 0 (Ni$^0$ d$^{10}$) − 3 (Bi$^{3-}$ s$^2$p$^6$) = 0], Bi impurity phases were detected in this composition. By artificially tuning the Zr vacancy concentration to ~12% (i.e., $Zr_{0.88}NiBi$), phase-pure samples can be repeatedly achieved. It should be noted that the XRD patterns for $Zr_{0.87}NiBi$ samples from two different batches showed contrasting phase purity despite the identical synthesis method used for each, where the sample from Batch I was found to be phase-pure while a tiny amount of Bi impurity was observed in the Batch II sample (Figure S1a). We speculate that the maximum Zr vacancy concentration for achieving stable phase purity is ~12%, and $Zr_{0.88}NiBi$ is thus designated for further discussion. Like typical 18-electron HHs, $Zr_{0.88}NiBi$ is crystallized in a cubic structure with space group $F\bar{4}3m$, in which Bi (purple) and Ni (red) atoms form a tetrahedrally bonded sublattice with Zr (green) atoms partially filling the octahedral voids around the tetrahedral sublattice, as schematically illustrated in Fig. 2b (the white sphere represents Zr vacancy, $V_{Zr}$).

Moreover, a JEM-ARM200F C$_s$-corrected transmission electron microscope (TEM) was employed to reveal the microstructure of $Zr_{0.88}NiBi$. As displayed in the low-magnification TEM image in Fig. 2c, visible string contrasts indicated by arrows are ubiquitous throughout the material, and these stem from the existence of a high density of twins. The high-angle annular dark-field scanning transmission electron microscopy (HAADF-STEM) image in Fig. 2d further illustrates the atomic configuration of this twinning structure. The shared twin plane between the matrix and the primary twin T1 is the ($\bar{1}$11) plane and the twin angle is ~55°, which corresponds well with the theoretical angle of 54.7° between the ($\bar{1}$11) and (002) planes. Interestingly, a secondary nanotwin T2 (delineated by blue dashed lines) is observed inside the primary twin T1. According to the atomistic dynamics of the self-activated hierarchical twinning behavior[25], the emission of Shockley partial dislocations from the intrinsic kinks on the twin boundaries will occur when the kink height is greater than five atomic layers, which in turn induces the nucleation and growth of secondary nanotwins. The unique twinning structure is also verified by the corresponding fast Fourier transform (FFT) (inset, Fig. 2d), in which the superposition of diffraction spots from the matrix, the primary twin, and the secondary nanotwin (marked by different colors) is evident. With its twofold twin, $Zr_{0.88}NiBi$ is distinguished among the family of HHs since reports of twinned HHs have been rare. The contributing role of high-energy ball-milling and annealing processes in the formation of twinning structures has been discussed for various materials[26,27], but it has not been reported for either ball-milled or annealed HHs. It should be noted that the "ball-milling + hot-pressing"-synthesized $Zr_{0.88}NiBi$ is not phase-pure, but the annihilation of the impurity phase can be observed

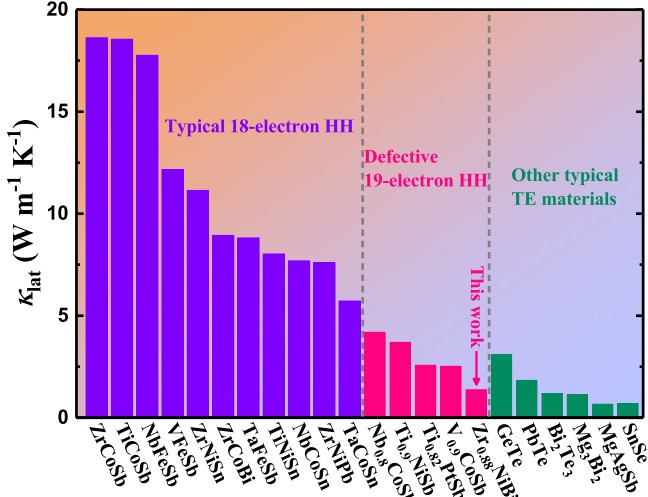

**Fig. 1** | Comparison of room-temperature lattice thermal conductivity among HHs and other typical thermoelectric (TE) materials[18,22,23].

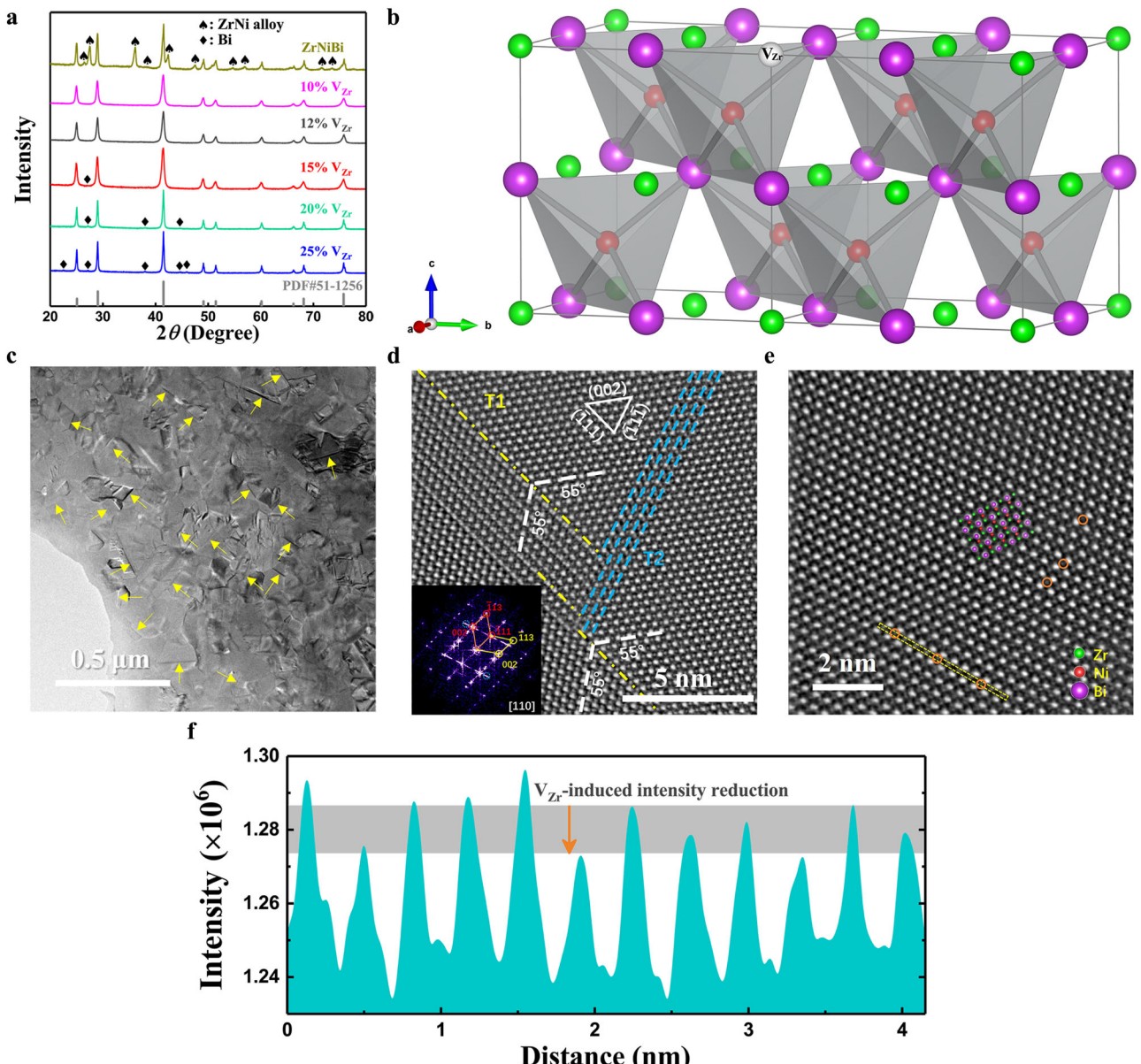

**Fig. 2 | Crystallographic properties of defective ZrNiBi. a** $V_{Zr}$-concentration-dependent XRD patterns for ZrNiBi-based compounds. **b** Crystal structure of phase-pure $Zr_{0.88}NiBi$. Green, red, purple, and white spheres represent Zr, Ni, Bi, and Zr vacancy, respectively. **c** Low-magnification TEM image of $Zr_{0.88}NiBi$. Twinning-structure-induced string contrasts are indicated by arrows. **d** Atomic-resolution HAADF-STEM image showing the twofold twinning structure of $Zr_{0.88}NiBi$. Yellow dash-dotted lines represent the boundary between the matrix and the primary twin T1 and blue dashed lines delineate the region of the secondary nanotwin T2. Inset: corresponding FFT diffraction pattern showing the superposition of diffraction spots from the matrix (red solid circles), the primary twin (yellow solid circles), and the secondary nanotwin (blue dotted circles). **e** HAADF-STEM image showing the atomic configuration of a single domain of $Zr_{0.88}NiBi$ and **f** the corresponding intensity profile within the 4a plane (region marked by the yellow dashed rectangle in **e**).

following an additional annealing process (Figure S1b). Such recrystallization behavior could correlate with thermally activated atomic diffusion, as well as with twinning-process-associated atomic rearrangement[28,29], where the formation of twins enables the existence of an interfacial network at a low energy state and thus stabilizes the structure[30].

In addition to the unique twinning configuration, the structure also accommodates a large concentration of cationic vacancy. Figure 2e presents the HAADF-STEM image of a single domain taken along the [110] direction. Crystallographically, the specific atom on each Wyckoff position 4a $(0, 0, 0)$ (i.e., Zr), 4b $(\frac{1}{2}, \frac{1}{2}, \frac{1}{2})$ (i.e., Bi), and 4c $(\frac{1}{4}, \frac{1}{4}, \frac{1}{4})$ (i.e., Ni) is visible without any overlapping along this direction. To verify the existence of Zr vacancies, the intensity profile within the 4a plane (region marked by a yellow dashed rectangle) was processed.

As displayed in Fig. 2f, the intensity reduction detected at different positions corresponding to the orange solid circles within the yellow dashed rectangle in Fig. 2e indicates that the Zr sites are not fully occupied. In contrast, only a slight intensity fluctuation of Bi atoms can be observed (Figure S2). In addition, the intrinsic Zr vacancies seem to be randomly distributed rather than exhibiting short-range order (SRO), which was discovered in other defective 19-electron HHs (e.g., $Nb_{0.8}CoSb$[22]). This can be ascertained from the dissimilar diffraction patterns for $Nb_{0.8}CoSb$ and $Zr_{0.88}NiBi$, in which the presence and absence, respectively, of a SROed-vacancy-induced diffuse band is observed. Xia et al. found that the SRO configuration favors a lower energy state[22], but the twinning configuration shows similar efficacy. In addition, stacking fault energy could be another coupling factor since SRO has a positive influence on it[31], while twins are more easily formed

in face-centered-cubic materials with low stacking fault energy[30]. As a result, competition between these two atomic configurations may occur. It should also be noted that the microstructure is highly sensitive to the synthesis process used, which allows for the possibility to finely control the atomic configuration, as well as the structure–property relationship, of defective HHs.

## Phononic dynamics

It is foreseeable that the intrinsic vacancies and twins in $Zr_{0.88}NiBi$ will endow it with strong phonon scattering. According to the Klemens model[32], point defects mainly scatter the high-frequency phonons and the scattering rate depends on the atomic mass fluctuation caused by the point defect. In principle, vacancy will produce much stronger fluctuation than substitutional impurity and thus acts as a more effective scattering source. Nonetheless, the phonon scattering effect of twinning structures remains a subject of debate. The results of some calculations and experiments have indicated that the contribution of twin boundaries to phonon scattering is significant, while other reports have argued that they have a weaker scattering effect than grain boundaries[33]. It should be noted that most twinned crystals have been reported with coherent twin boundaries, and such coherence is suspected to induce minor thermal boundary resistance. In $Zr_{0.88}NiBi$, the formation of the secondary nanotwin originates from the defect (kink) on the twin boundary, implying that the compound's lattice coherence has been interrupted. It is thus reasonable that the twofold twinning configuration also contributes to effective phonon scattering.

More intriguingly, $Zr_{0.88}NiBi$ possesses much lower $\kappa_{lat}$ than $Nb_{0.8}CoSb$ (Fig. 3a) although its vacancy concentration is lower, indicating that there are other underlying mechanisms responsible for its extraordinarily low $\kappa_{lat}$. The kinetic theory provides a direct correlation between a solid material's $\kappa_{lat}$ and its phonon relaxation time (determined by scattering) and group velocity[34]. Since the phonon group velocity in the low-frequency region can be approximated by the sound velocity ($v_s$), the measured $v_s$ values of various HHs were compared for further discussion[16,35–38]. In a ball-and-spring lattice, $v_s$ is governed by the restoring force ($F$) and the average mass ($\bar{M}$) as $v_s \sim (\frac{F}{\bar{M}})^{1/2}$. As shown in Fig. 3b, there is no doubt that heavy-element-containing HHs exhibit lower $v_s$. For instance, the $v_s$ values of $Zr_{0.88}NiBi$ and $Nb_{0.8}CoSb$ are 2500 m s$^{-1}$ and 3313 m s$^{-1}$, respectively. In addition, the relativistic contraction of the Bi-6$s$ orbital enables inert chemical bonds[36] (i.e., a global weak interatomic force). It should also be noted that, regardless of their similar $\bar{M}$ or the nature of Bi, $Zr_{0.88}NiBi$ is distinguished among the heavy-element-containing HHs by its anomalously low $v_s$. We assumed a much weaker interatomic force in $Zr_{0.88}NiBi$ and then calculated its electron localization function (ELF) and bond lengths to investigate its bonding characteristics ($2 \times 2 \times 2$ supercell with removal of one Zr atom corresponding to $Zr_{0.875}NiBi$, hereafter approximated as $Zr_{0.88}NiBi$). The ELF value ranges from 0 to 1, where ELF = 1 indicates complete localization of electrons. Figure 3c, d illustrates the projected two-dimensional ELFs along the $(01\bar{1})$ plane for $Zr_{0.88}NiBi$ and ZrCoBi, respectively, in both of which thin gray circle-like contour lines denote regions with the ELF value of 0.8, thus revealing strong electron localization in the Bi atoms for both materials. This phenomenon is similar to that observed in $RhBi_4$, which was attributed to the material's lone pairs of Bi atoms[39]. More intriguingly, asymmetric distribution of the electron cloud in the Bi atoms that adjoin Zr vacancies is observed in $Zr_{0.88}NiBi$, as evidenced by the existence of regions with the higher ELF value of 0.85 (denoted by the thick black crescent-like contour lines in Fig. 3c) in the direction of the vacancies. As a result, a local structural distortion occurs and is accompanied by diverse bond lengths ($d_{Ni-Bi} = 2.67$–2.78 Å and $d_{Zr-Ni} = 2.68$–2.71 Å), in contrast to ZrCoBi, which shows identical Co–Bi and Zr–Co bond lengths (~2.70 Å). Such diversity in bond lengths permits large and anisotropic atomic displacement, which is a typical indicator of soft bonds. Atomic displacement parameters (ADPs) will be discussed later. Both the appearance of asymmetric electron localization in the vacancy-neighboring Bi atoms in $Zr_{0.88}NiBi$ and the slight changes in its Zr–Ni bond lengths indicate the local softening of chemical bonds in this case, and this can also be ascertained from the results of ADP analysis discussed below. Evidently, the nature of Bi and the existence of Zr vacancy are together responsible for the hierarchical chemical bonds (coexistence of global and local soft bonds) in $Zr_{0.88}NiBi$, which can disrupt phonon propagation and result in much lower $v_s$.

To further discern the fantastic phonon dynamics of $Zr_{0.88}NiBi$, its phonon dispersion, phonon density of states (PDOS), and temperature-dependent ADPs were studied (Fig. 3e). First, no imaginary frequency was observed in the phonon dispersion of $Zr_{0.88}NiBi$, implying its dynamical stability. In contrast to corresponding results for ZrCoBi (Fig. 3f) and phonon dispersion results for the supercell of $Zr_8Co_8Bi_8$ (Figure S3), the flattened phonon dispersion and large ADPs of $Zr_{0.88}NiBi$ reinforce our finding of its soft bonding nature since the reduced overall group velocity can be directly visualized by the flattened phonon dispersion and the loosely bound atoms tend to vibrate with larger displacement. The PDOS of $Zr_{0.88}NiBi$ shows that its lattice vibration with a frequency below ~3 THz is mainly dominated by Bi atoms and that the cutoff frequency of its acoustic phonons is ~2.2 THz, which is significantly lower than that for ZrCoBi (approaching 4 THz). Moreover, differing from ZrCoBi with a large acoustic-optical gap, $Zr_{0.88}NiBi$ features low-lying optical phonons (1.5–3 THz) that are strongly hybridized with acoustic branches (in the frequency range between 1.87 and 2.22 THz). We expect that these low-lying optical phonons originate from the compound's local soft bonds, and an analogous phenomenon was also reported for α-MgAgSb[40]. Importantly, these soft optical modes with nearly vanishing group velocities show a non-propagating-like behavior while enabling additional scattering for heat-carrying acoustic phonons. The prevailing viewpoint is that thermal resistance results from the lowest-order three-phonon scattering, which gives rise to a $T^{-1}$ dependence of $\kappa_{lat}$[34]. However, a higher-order phonon scattering process considering optical phonons has gained increasing attention and its adequacy has also been proved in other materials such as boron arsenide[41–43] and hybrid perovskites[44,45].

It should be noted that there are visible fluctuations in the ADPs for each atom in the supercell of $Zr_{0.88}NiBi$, as shown in Fig. 3e with details along the three Cartesian directions plotted in Figure S4, which is attributed to the diversity of bond lengths. The ratios of the largest to smallest ADPs for Zr, Ni, and Bi are 1.24:1, 1.12:1, and 1.65:1, respectively, at room temperature. Moreover, the smallest ADP for Bi in $Zr_{0.88}NiBi$ shows only a minor difference compared to the ADP for Bi in ZrCoBi, indicating the similar vibrating modes in these two materials governed by a global soft bond. The fact that the largest and second-largest ADPs respectively correspond to Bi5 and Bi4 (both along the y-axis), which both show an asymmetrically distributed electron cloud adjacent to a Zr vacancy (see Fig. 3c), attests to the existence of a vacancy-induced much softer local chemical bond. In addition, decent lattice anharmonicity in $Zr_{0.88}NiBi$ is evidenced by its anisotropic ADPs, especially at ambient temperature. Such anisotropic behavior is usually absent in highly symmetrical HHs (e.g., identical ADPs along the three Cartesian directions for ZrCoBi) and is also correlated to the local soft bonds in the case of $Zr_{0.88}NiBi$. Therefore, our in-depth analysis of vacancy-mediated anomalous phononic transport in $Zr_{0.88}NiBi$ successfully elucidates the origin of its intrinsically low $\kappa_{lat}$, which is attributed to the multiple phonon scattering processes in the compound and its global and local soft bonding environment.

## Electronic properties

Given the multifunctionality of vacancy in tailoring phononic properties, as well as the potential for high thermoelectric performance, the

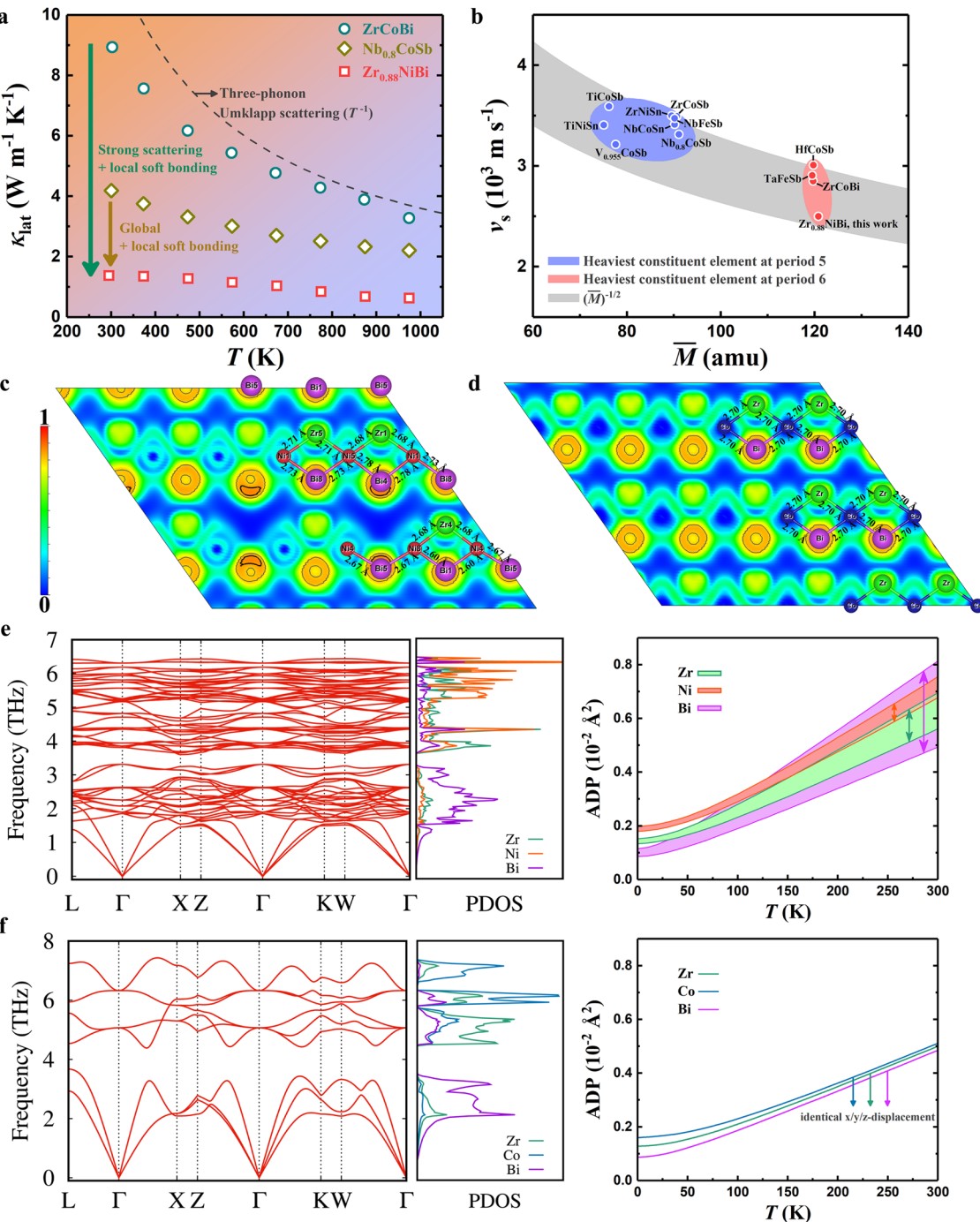

**Fig. 3 | Phononic properties of defective ZrNiBi. a** Comparison of temperature-dependent lattice thermal conductivity among $Zr_{0.88}NiBi$, ZrCoBi, and $Nb_{0.8}CoSb$. **b** Experimentally determined sound velocity of various HHs with respect to their average atomic mass[16,35–38]. The gray shaded region shows the trend of $v_s \sim (\overline{M})^{-1/2}$ and the corresponding data of possible HHs are from the *AFLOWLIB.org* database[65] (see details in the Supplemental Material). Calculated two-dimensional ELFs along the (01$\bar{1}$) plane together with corresponding bond lengths for **c** $Zr_{0.88}NiBi$ and **d** ZrCoBi. Thin gray circle-like and thick black crescent-like contour lines correspond to the ELF values of 0.8 and 0.85, respectively. Calculated phonon dispersion, PDOS, and ADPs for **e** $Zr_{0.88}NiBi$ and **f** ZrCoBi. Ranges of ADPs for all atoms in the supercell of $Zr_{0.88}NiBi$ (see Figure S4) are shown in (**e**).

electronic properties of $Zr_{0.88}NiBi$ deserve further attention. The electron-sufficient configuration of $Zr_{0.88}NiBi$ results in n-type self-doping behavior, which can also be identified by the negative Seebeck coefficient (*S*). As seen in the unfolded electronic band structure of $Zr_{0.88}NiBi$ (Fig. 4a), the Fermi level lies well above the conductive band minimum (CBM) with an electron concentration of ~$9.89 \times 10^{21}$ cm$^{-3}$ and the CBM locates at the X point with three-fold valley degeneracy, which is analogous to most 18-electron HHs. By plotting the Pisarenko relation as shown in Fig. 4b, the modeled density of states (DOS)

effective mass is ~6.0 $m_e$ (details for the single parabolic band model are provided in the Supplemental Material). It should be noted that the CBM of a HH is primarily governed by the interaction between the *d*-orbitals of two transition metals[46] (e.g., Zr and Ni in the compound studied here). This orbital configuration should give rise to a similar conductive band structure among different compounds when the two transition metals are unchanged, such as the comparable DOS effective mass values of ZrNiSn and ZrNiPb (~3.0 $m_e$)[47,48]. In addition, ZrCoSb and ZrCoBi also display identical behavior[49,50], further

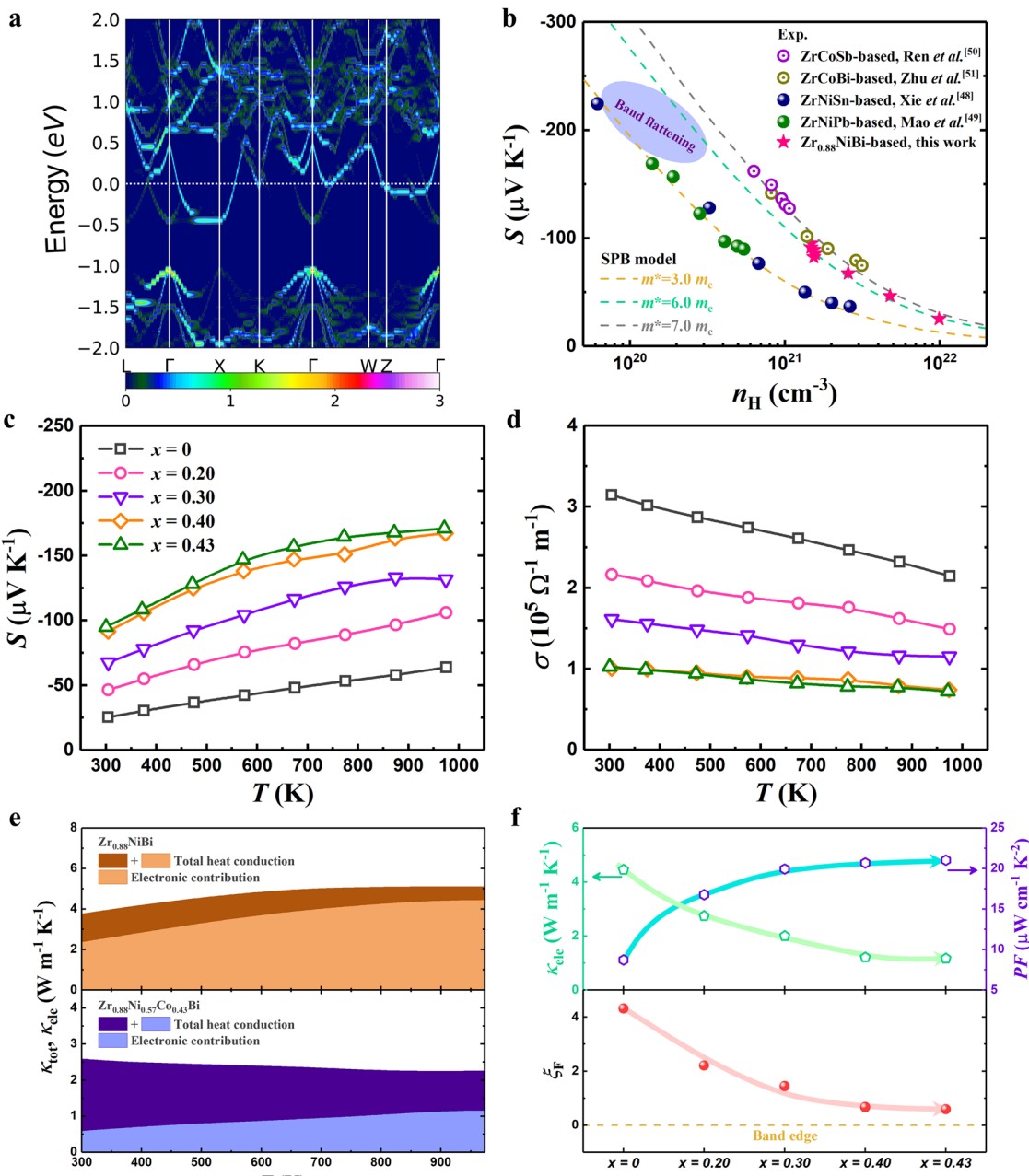

**Fig. 4 | Electronic properties of defective ZrNiBi. a** Calculated unfolded electronic band structure of $Zr_{0.88}NiBi$. **b** Pisarenko relation of $Zr_{0.88}NiBi$ in comparison to that of ZrNiSn[47], ZrNiPb[48], ZrCoSb[49], and ZrCoBi[50]. Symbols: experimental data. Dashed lines: obtained from the SPB model. Temperature-dependent **c** Seebeck coefficient and **d** electrical conductivity of $Zr_{0.88}Ni_{1-x}Co_xBi$. **e** Comparison of the electronic contribution to total heat conduction between $Zr_{0.88}NiBi$ (top panel) and $Zr_{0.88}Ni_{0.57}Co_{0.43}Bi$ (bottom panel). **f** Composition-dependent electronic thermal conductivity and power factor (top panel) and reduced Fermi energy (bottom panel) for $Zr_{0.88}Ni_{1-x}Co_xBi$.

confirming the negligible influence of the main group element on the CBM. The larger $m^*$ of $Zr_{0.88}NiBi$ thus seems counterintuitive, but the flat band near the CBM along the Γ and X directions that exists in $Zr_{0.88}NiBi$ is not observed in either ZrNiSn or ZrNiPb. This flat band probably arises from the localized orbital around the vacancy, resembling $HfO_2$, in which the vacancy causes a local spin state and unexpected ferromagnetism[6]. Moreover, based on the theory of topological quantum chemistry[51], the nature of symmetry-enforced Fermi degeneracy at high-symmetry points qualifies this material as a type of topological electronic material, analogous to YPdBi documented in ref. 52. We thus speculate that this material could attract considerable attention for condensed matter physics research due to its exotic band structure. We further estimated the mean free path of

an electron at ~6.33 Å based on the formula $mfp = \sqrt{2E_F m_b^*}\frac{\mu}{e}$, where $E_F$, $m_b^*$, $\mu$, and $e$ are the Fermi energy, effective band mass ($m^* = N_V^{2/3}m_b^*$), mobility, and elementary electron charge, respectively. The $mfp$ of $Zr_{0.88}NiBi$ is comparable to its lattice constant (~6.19 Å), suggesting that its electron transport has reached the Ioffe–Regel limit[53], which is reasonable given its unconventionally defective structure. This also implies that the carrier filtering effect from twin boundaries could be negligible since electrons are scattered before this effect occurs.

Nonetheless, a deep Fermi level location is unfavorable to good thermoelectric performance. Hong et al. specified that the optimal position of the Fermi level is near the band edge[54]. To downshift the Fermi level in the $Zr_{0.88}NiBi$-based compound, acceptor cobalt was thus doped for electron capture. The temperature-dependent $S$ of

$Zr_{0.88}Ni_{1-x}Co_xBi$ is presented in Fig. 4c. In accordance with the inverse correlation between $S$ and carrier concentration, $|S|$ increases with increasing doping concentration ($x$), suggesting that cobalt provides efficient electron capture. As a result, the electron concentration is reduced to ~$1.48 \times 10^{21}$ cm$^{-3}$ for $Zr_{0.88}Ni_{0.57}Co_{0.43}Bi$. Electrical conductivity ($\sigma$) decreases with increasing $x$, also in accordance with the reduced electron concentration (Fig. 4d). Since electronic thermal conductivity ($\kappa_{ele}$) is determined by the carrier concentration as well, high $\kappa_{ele}$ is a major drawback of $Zr_{0.88}NiBi$ that can be addressed via the efficient electron capture provided by cobalt doping. As shown in the top panel of Fig. 4e, the contribution to total thermal conductivity ($\kappa_{tot}$) from electrons is ~63% at room temperature and, following the Wiedemann-Franz law, becomes more significant with increasing temperature. On the other hand, the electronic contribution is significantly suppressed in $Zr_{0.88}Ni_{0.57}Co_{0.43}Bi$ (bottom panel, Fig. 4e), such that phonons have the dominant role in heat conduction (the phononic contribution can be found in Figure S5). Consequently, cobalt doping enables synergistic optimization of both $\kappa_{ele}$ and power factor ($PF$, the product $S^2\sigma$), which is a unique advantage over the typical 18-electron HHs. Most 18-electron HHs are intrinsic semiconductors and optimization of their performance thus relies on an elevated carrier concentration that will inevitably lead to a monotonically increased $\kappa_{ele}$. The countertrend behavior of the compound studied here is illustrated in the top panel of Fig. 4f. We further determined the composition-dependent reduced Fermi energy ($\xi_F = \frac{E_F}{k_B T}$, where $k_B$ is the Boltzmann constant), and the results are shown in the bottom panel of Fig. 4f. The band-edge-approaching $\xi_F$ observed for $Zr_{0.88}Ni_{0.57}Co_{0.43}Bi$ is in line with expectations given the corresponding improvement in thermoelectric performance.

## Figure of merit

Benefitting from the efficient electron capture discussed above, the peak $ZT$ of $Zr_{0.88}Ni_{0.57}Co_{0.43}Bi$ is five times that of the undoped $Zr_{0.88}NiBi$ (Fig. 5a). By further alloying isovalent antimony at the bismuth site, a peak $ZT$ of ~1.0 at 973 K is obtained for $Zr_{0.88}Ni_{0.57}Co_{0.43}Bi_{0.9}Sb_{0.1}$ (the thermoelectric properties of $Zr_{0.88}Ni_{0.57}Co_{0.43}Bi_{1-y}Sb_y$ are shown in Figure S6; additionally, the phase purity of all $Zr_{0.88}Ni_{1-x}Co_xBi_{1-y}Sb_y$ samples studied is shown in Figure S1c). The performance advantage of this material over other typical n-type HHs is evidenced by the comparisons shown in Fig. 5b, c[16,18,35,38,49,50,55–58]. Although the archetypal ZrNiSn-based HH exhibits similar $ZT$ values at lower temperatures, the bipolar transport effect causes its performance to degrade at high temperature, making it incompatible with p-type HHs working at 973 K. In comparison to the recently reported n-type ZrCoBi-based HH with high performance at 973 K, an overall enhanced $ZT$ can be observed in the material studied here. Consequently, $Zr_{0.88}Ni_{0.57}Co_{0.43}Bi_{0.9}Sb_{0.1}$ exhibits a decent average $ZT$ (over the range of 300–973 K), outperforming various defective 19-electron and typical 18-electron HHs.

## Discussion

Given that defects have been regarded as an additional degree of freedom for boosting thermoelectric performance, studies of the vacancy-abundant half-Heuslers have the potential to reshape the development of high-temperature thermoelectric materials. Here we modeled the figure of merit of the defective ZrNiBi at 973 K. When the electron transport of defective ZrNiBi reaches the Ioffe–Regel limit (electron mean free path ~$a$, where $a$ denotes the lattice constant)[53] and its phonon transport follows the Cahill model (minimal lattice thermal conductivity ~0.66 W m$^{-1}$ K$^{-1}$)[59], its figure of merit exhibits an impressive value of ~2.2 at the electron concentration of $4$–$5 \times 10^{20}$ cm$^{-3}$ (Figure S7a). It should be noted that our transmission electron microscope results revealed a random distribution of vacancies in $Zr_{0.88}NiBi$, in contrast to some typical defective 19-electron half-Heuslers that have been reported to show short-range order among their vacancies (e.g.,

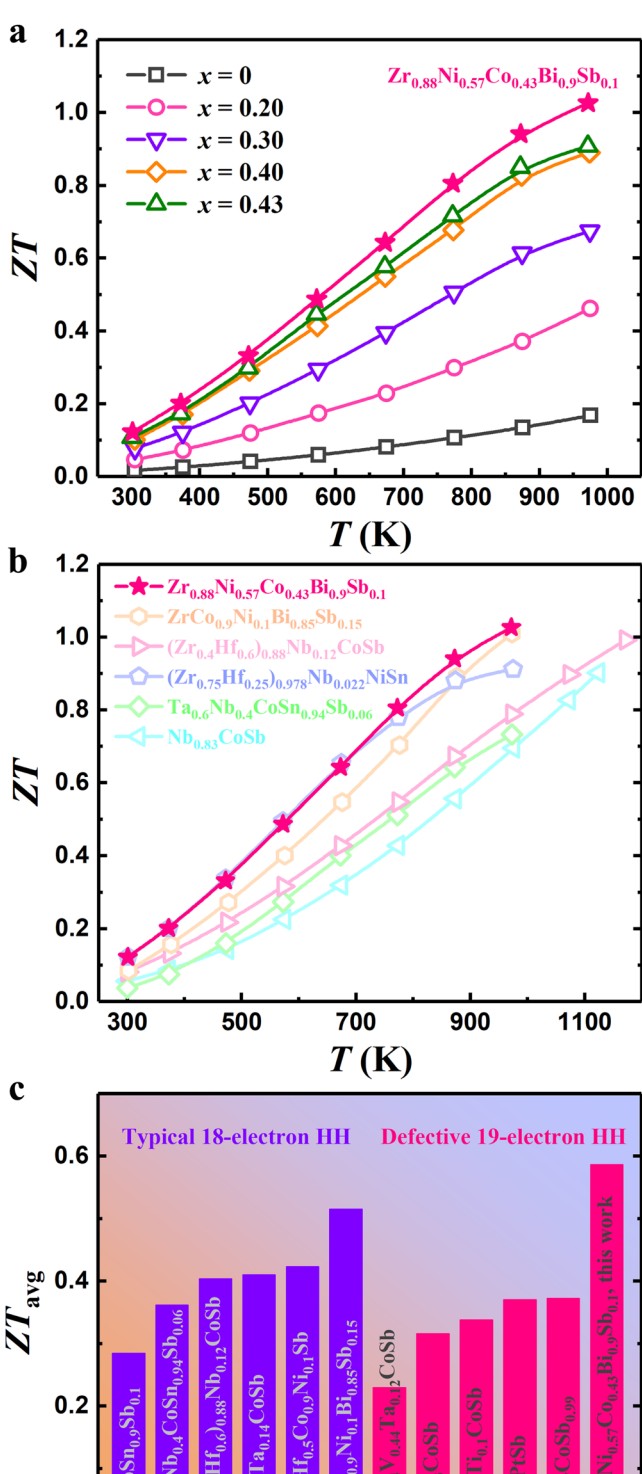

**Fig. 5 | Thermoelectric properties of defective ZrNiBi. a** Temperature-dependent $ZT$ of $Zr_{0.88}Ni_{1-x}Co_xBi$ and $Zr_{0.88}Ni_{0.57}Co_{0.43}Bi_{0.9}Sb_{0.1}$. Comparisons of **b** temperature-dependent $ZT$ and **c** average $ZT$ between $Zr_{0.88}Ni_{0.57}Co_{0.43}Bi_{0.9}Sb_{0.1}$ and various typical n-type HHs[16,18,35,38,49,50,55–58].

$Nb_{0.8}CoSb$[22]). It is thus natural to examine whether the performance of defective ZrNiBi could be enhanced if its degree of vacancy order can be improved. However, the existence of vacancy pairs and vacancy triplets complicates the short-range order in defective 19-electron half-Heusler systems[22]. Since long-range order is an extreme case of vacancy order,

we assumed the existence of long-range-ordered vacancies in $Zr_{0.88}NiBi$ (where the shortest inter-vacancy distance of $\sqrt{2}a$ is assigned to the electron mean free path) and, correspondingly, the material's figure of merit as a function of both lattice thermal conductivity and electron concentration was plotted in Figure S7b. In comparison to the framework of the Ioffe–Regel condition, the optimal electron concentration is shifted to a lower range and the material exhibits better performance, e.g., its peak figure of merit is enhanced to ~2.7 as the lattice thermal conductivity decreases to the value estimated by the Cahill model. It must be noted that a system with only long-range-ordered vacancies is hardly achievable based on experimental results, so the trend found from our model sets an upper bound for the thermoelectric performance of defective ZrNiBi while the degree of vacancy order can serve as a control for its figure of merit. In fact, current optimization approaches have resulted in a peak figure of merit of ~1.0 for a $Zr_{0.88}NiBi$-based material at a lattice thermal conductivity of ~0.9 W m$^{-1}$ K$^{-1}$ and an electron concentration of ~1.5 × 10$^{21}$ cm$^{-3}$. To fulfill its thermoelectric potential, delicate phonon and electron concentration engineering (toward lower values for both lattice thermal conductivity and electron concentration) is necessary. In addition, vacancy manipulation for ordered distribution enables further enhancement of performance as depicted in Figure S7c.

In summary, an emerging defective ZrNiBi-based compound with distinctive structural characteristics and anomalous transport properties that distinguish it among the half-Heuslers is introduced. Its randomly distributed cationic vacancies and twofold twins were directly visualized using real-space imaging. Phonon dynamics analysis demonstrated that vacancy-induced point defect scattering, low-lying optical phonons, anisotropic vibrations, and local soft bonds, going beyond the conventional vacancy scattering model, together correlate highly with the compound's extraordinarily low lattice thermal conductivity (~1.4 W m$^{-1}$ K$^{-1}$ at room temperature). Regarding its electronic properties, experimental Pisarenko relations and first-principles calculations confirmed the existence of a flat band near the conductive band minimum that is also vacancy related and has been rarely reported in other half-Heuslers. Together with the symmetry-enforced Fermi degeneracy at high-symmetry points, the exotic band structure makes defective ZrNiBi quite appealing for condensed matter physics research. With the benefits of vacancy-mediated anomalous transport properties and optimized downshifting of the Fermi level to the band edge, significantly enhanced thermoelectric performance was achieved in a defective ZrNiBi-based material, outperforming various typical n-type half-Heuslers. We expect that the discovery of this promising thermoelectric material could further advance the potential of thermoelectric half-Heuslers and that our findings of vacancy-mediated anomalous transport properties will trigger in-depth research on other fascinating defect-driven phenomena.

## Methods

### Material synthesis

The phase-pure ZrNiBi-based compounds were synthesized by ball-milling, hot-pressing, and annealing processes with high-purity raw elements (Zr sponges, 99.2%; Ni powder, 99.8%; Bi pieces, 99.99%; Co powder, 99.8%; and Sb shots, 99.8%). The raw elements were stoichiometrically weighed and then loaded into an argon-filled stainless steel jar for the subsequent ball-milling process. After 20 h of ball-milling, each obtained nanoscale powder sample was hot-pressed at ~1000 K with a holding time of 2 min. Each resulting solidified disk was sealed in a quartz tube and then annealed at 973 K for 40 h. The heating rates for hot-pressing and annealing were ~150 K/min and 3.8 K/min, respectively.

### Structure characterization

X-ray diffraction (XRD, Rigaku SmartLab diffractometer) was performed to identify the phase composition of each sample.

Transmission electron microscopy (TEM) and high-angle annular dark-field scanning transmission electron microscopy (HAADF-STEM) investigations were carried out using a double-$C_s$ corrected JEM-ARM200F equipped with a cold field emission gun (FEG) source. Specimens used for TEM observation were prepared by traditional mechanical polishing, dimpling, and subsequent ion milling with a liquid nitrogen stage. All the HAADF images in this study were Fourier-filtered to reduce the influence of random noise.

### Transport measurement

The Seebeck coefficient ($S$) and electrical conductivity ($\sigma$) of each sample were simultaneously measured using a commercial system (ZEM-3, ULVAC). Hall carrier concentration ($n_H$) was obtained using a physical properties measurement system (PPMS, Quantum Design) under a magnetic field of ±7 T and electrical current of 8 mA. Thermal conductivity ($\kappa_{tot}$) was calculated based on $\kappa_{tot} = DC_P\rho$, where $D$ is thermal diffusivity measured by laser flash (LFA 457, Netzsch), $C_P$ is the specific heat obtained by the Dulong-Petit law, and $\rho$ is mass density determined by an Archimedes' method. Electronic thermal conductivity ($\kappa_{ele}$) was calculated based on $\kappa_{ele} = L\sigma T$, where $L$ is the Lorenz number deduced by the single parabolic band model and $T$ is the absolute temperature. Lattice thermal conductivity ($\kappa_{lat}$) was determined by subtracting the $\kappa_{ele}$ from $\kappa_{tot}$. Sound velocity ($v_s$) studies were carried out using a RITEC RAM-5000 Advanced Ultrasonic Measurement System, which incorporates a pulse-echo technique for time propagation measurements. Piezoelectric transducers (Olympus NDT, Inc.) operating at a frequency of 10 MHz were used to generate and receive longitudinal and shear ultrasonic bulk waves. Propylene glycol and shear wave couplant (SWC) (both from Olympus NDT Inc.) were used as couplant materials for longitudinal and shear modes, respectively. The time resolution achieved in our experiments was better than 0.2 ns.

### Theoretical calculation

First-principles density functional theory (DFT) calculations were carried out using the Vienna ab initio Simulation Package (VASP) with the projector augmented wave (PAW) scheme[60]. The Perdew, Burke, and Ernzerhof (PBE) electronic exchange correlation functional was used under a gradient-generalized approximation (GGA)[61]. We used a constant energy cutoff of 500 eV to truncate the plane-wave basis and a roughly constant density of k-points (30 Å$^3$) of Monkhorst-Pack[62] point meshes to sample the Brillouin zones. All structures were relaxed with respect to all the forces, and components of the tensors were below 0.01 eV/Å and 0.2 kbar, respectively. A 2 × 2 × 2 supercell with 23 atoms following removal of one Zr atom (i.e., $Zr_7Ni_8Bi_8$) was used to calculate the electronic structure of $Zr_{0.875}NiBi$. To determine the band structure, the band unfolding method (BandUP code)[63] was used to unfold the band structure of the supercell into its primitive Brillouin zone. Phonon dispersions of $Zr_{0.875}NiBi$ and ZrCoBi were calculated using 2 × 2 × 2 supercells with 184 atoms and 3 × 3 × 3 supercells with 81 atoms, respectively, by the finite displacement method implemented in the Phonopy package[64].

## Data availability

All data are available in the main text and the Supplementary Materials.

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

## Acknowledgements

Z.M.W. acknowledges financial support from the National Key Research and Development Program of China (2019YFB2203400) and the "111 Project" (B20030). W.Y.R. acknowledges financial support from the Sichuan Science and Technology Program (2023NSFSC0970). J.W. acknowledges financial support from the Innovation Group Project of Sichuan Province (20CXTD0090). R.H. acknowledges financial support from the Deutsche Forschungsgemeinschaft (DFG, 453261231). J.v.d.B. acknowledges the Deutsche Forschungsgemeinschaft (DFG) for support through the Wrüzburg-Dresden Cluster of Excellence on Complexity and Topology in Quantum Matter ct.qmat (EXC 2147, Project No. 39085490) and the Collaborative Research Center SFB 1143 (Project No. 247310070). C.W.C. acknowledges financial support from the U.S. Air Force Office of Scientific Research Grants FA9550-15-1-0236 and FA9550-20-1-0068, the T.L.L. Temple Foundation, and the John J. and Rebecca Moores Endowment. S.G. thanks Oleg Janson for fruitful discussions and Ulrike Nitzsche at IFW Dresden for technical assistance. We also thank Troy Christensen at the University of Houston for revising the paper and Professor David J. Singh at University of Missouri for discussions.

## Author contributions

W.Y.R., Z.M.W., and Z.F.R. designed this work. W.Y.R. and S.W.S. synthesized the samples. W.Y.R., R.H., L.Z.D., and A.S. conducted the transport property measurements. W.Y.R., W.H.X., and Y.M.W. performed the structure characterizations. S.P.G. and J.v.d.B. did the theoretical calculations. K.N., J.W., and C.W.C. helped with the transport property analysis. G.H.G., S.C., and Y.M.H. helped with the TEM result analysis. W.Y.R. and Z.F.R. wrote the manuscript, and all authors edited it.

## Competing interests

The authors declare no competing interests.
