## [Peer Review File · Nature Communications]

Vacancy-mediated anomalous phononic and electronic transport in defective half-Heusler ZrNiBiREVIEWER COMMENTS

Reviewer #1 (Remarks to the Author):

In this work, the authors systematically investigated the electrical and thermal transport properties of defective ZrNiBi. Randomly distributed zirconium vacancy and twofold twinning were revealed using high-angle annular dark-field scanning transmission electron microscopy. An exceptionally low κ_{lat} of $\sim 1.4 \text{ W m}^{-1} \text{ K}^{-1}$ at room temperature was obtained in defective ZrNiBi. The electronic band structure is modified by the cationic vacancy so that a flat band was observed near the conductive band minimum. The peak ZT of $\text{Zr}_{0.88}\text{Ni}_{0.57}\text{Co}_{0.43}\text{Bi}$ is five times that of the undoped $\text{Zr}_{0.88}\text{NiBi}$. By further alloying isovalent antimony at the bismuth site, a peak ZT of ~ 1.0 at 973 K is obtained for $\text{Zr}_{0.88}\text{Ni}_{0.57}\text{Co}_{0.43}\text{Bi}_{0.9}$. This work is interesting. I recommend it to be published after minor revisions.

1. What about the minimum Zr-deficiency for achieving phase-pure Zn-Ni-Bi?
2. The discussion about the relation between the twins and the annihilation of impurity phase is obscuring. What are the impurity phases? Why the annihilation of the impurity phases has relation with the twins?
3. $\text{Zr}_{0.88}\text{NiBi}$ and ZrCoBi have similar average atomic mass. Why ZrCoBi has higher sound velocity than $\text{Zr}_{0.88}\text{NiBi}$?
4. Have the authors calculated the phonon dispersion of ZrNiBi without Zr-vacancy? In principle, ZrNiBi is not stable and imaginary frequency might appear in the phonon dispersion.
5. In Fig. 3, it is better to use the same supercell to calculate the phonon dispersions of $\text{Zr}_{0.88}\text{NiBi}$ and ZrCoBi .
6. In Fig. 3E, why the Ni atoms have larger ADP than Zr and Bi atoms?
7. In Fig. 4b, the data of ZrCoBi should be also included for comparison.
8. I suggest the authors to comment how to further optimize the TE performance of defective half-Heusler alloys.

Reviewer #2 (Remarks to the Author):

The authors studied how vacancy affected the thermoelectric properties of defective half-Heusler ZrNiBi . I appreciate the detailed study the authors carried out by combining experiments and computation. It would be an important piece of work for the community. I have the following questions/comments for the authors to consider:

1. The biggest question I have is what really is “anomalous.” I feel all the results were expected. The theoretical treatment of vacancy has been challenging, so people often used the empirical equation to treat it as a phonon scattering center with six times the mass of the original atom. But it doesn't mean that people are unaware of the effects of softening bonds, lowering group velocities, etc. For example, using the virtual crystal model for alloys can lead to overestimating the phonon lifetimes if the local force constants variations were not included [Journal of Physics: Condensed Matter 27 (37), 375403]. In this sense, I would appreciate it if the authors could better cultivate/support their claim of “anomalous.” Or maybe it is better to remove “anomalous” and claim a comprehensive understanding of
2. In the “Theoretical calculation” section, “A $2 \times 2 \times 2$ supercell with 23 atoms following removal of one Zr atom was used to calculate the electronic structure of $\text{Zr}_{0.875}\text{NiBi}$.” Were 23 atoms the number of atoms in the unit cell? Please clarify.
3. Can the authors comment on how much higher they expect ZT to reach if vacancies are optimized to promote the thermoelectric properties?
4. I wasn't sure why the discussion about “twinning” was relevant to the vacancy.
5. The phonon dispersion features of no acoustic-optical bandgap are quite similar to hybrid perovskites that possess ultralow thermal conductivity. It would be nice to include or

comment on that. [Physical review letters 123 (15), 155901; Nano Letters 21 (9), 3708-3714].

Reviewer #3 (Remarks to the Author):

The present manuscript reported on a defective ZrNiBi HH alloy with an ultralow lattice thermal conductivity and a high ZT value. The low kappa lattice was attributed to the soft chemical bonds, acoustic-optical phonon coupling, and point defect scattering, etc. induced by the presence of Zr vacancies. A vacancy-induced flat band was also observed, which leads to much-improved density-of-states effective mass and power factor. By introducing proper dopants to tune the carrier concentration, a peak ZT of 1.0 at 973 K is obtained. The conclusions are supported by comprehensive structural characterizations and theoretical calculations. This work reveals the multifunctional properties of vacancies in mediating anomalous phononic and electronic transports. It could provide a new paradigm for property modifications by controlling vacancies. Thus, I recommend the publication of this work in Nature Communications with minor revisions. My specific comments are listed below.

1. The authors emphasized the observation of twinning structures but did not discuss the impact of twinning on the thermoelectric properties. It has been reported in many studies that the twin structure can influence the electronic band structure and phonon scattering.
2. The local soft chemical bonding was derived from the electron localization from ELF values. I do not understand the correlations between a large ELF and a soft chemical bond. This seems to contradict the intuition that strong electron localization is generally obtained in stiff bonds such as covalent and ionic bonds. In contrast, the soft metallic and metavalent bonds often show a strong electron delocalization.
3. The authors described the coupling of low-lying optical phonons and acoustic phonons in sample Zr_{0.88}NiBi. Yet, it is not very convenient to observe this phenomenon in the calculated phonon spectrum. It would be helpful if the corresponding phonon branches can be indicated by different colors or by arrows. I do not see obvious TO softening at the Γ point compared with the typical phenomenon observed in anharmonic PbTe.

Response to reviewers' comments

Response to Reviewer 1:

Reviewer #1: In this work, the authors systematically investigated the electrical and thermal transport properties of defective ZrNiBi. Randomly distributed zirconium vacancy and twofold twinning were revealed using high-angle annular dark-field scanning transmission electron microscopy. An exceptionally low κ_{lat} of $\sim 1.4 \text{ W m}^{-1} \text{ K}^{-1}$ at room temperature was obtained in defective ZrNiBi. The electronic band structure is modified by the cationic vacancy so that a flat band was observed near the conductive band minimum. The peak ZT of $\text{Zr}_{0.88}\text{Ni}_{0.57}\text{Co}_{0.43}\text{Bi}$ is five times that of the undoped $\text{Zr}_{0.88}\text{NiBi}$. By further alloying isovalent antimony at the bismuth site, a peak ZT of ~ 1.0 at 973 K is obtained for $\text{Zr}_{0.88}\text{Ni}_{0.57}\text{Co}_{0.43}\text{Bi}_{0.9}$. This work is interesting. I recommend it to be published after minor revisions.

Response: We greatly appreciate the reviewer's time and effort in reading our manuscript, understanding its value, and providing valuable suggestions. Following are our point-by-point responses to the reviewer's constructive comments.

Comment 1: What about the minimum Zr-deficiency for achieving phase-pure Zn-Ni-Bi?

Response: We thank the reviewer for the comment. Vacancy concentration is crucial to the phase purity of this material. To determine the maximum Zr deficiency that will ensure stable phase purity, we further compared the XRD patterns for $\text{Zr}_{0.88}\text{NiBi}$ and $\text{Zr}_{0.87}\text{NiBi}$ from different batches that were each prepared using an identical synthesis method, as shown in Figure R1. Differing from stable $\text{Zr}_{0.88}\text{NiBi}$, $\text{Zr}_{0.87}\text{NiBi}$ from Batch I is phase-pure while a tiny amount of Bi impurity is observed in the sample from Batch II despite using an identical synthesis method for each. This indicates 0.88 is the critical value. The maximum Zr vacancy concentration for achieving phase-pure Zr-Ni-Bi is approximately 12% based on the above results.

For clarification, the following has been added on page 4 of the revised manuscript: "It should be noted that the XRD patterns for $\text{Zr}_{0.87}\text{NiBi}$ samples from two different batches showed contrasting phase purity despite the identical synthesis method used for each, where the sample from Batch I was found to be phase-pure while a tiny amount of Bi impurity was observed in the Batch II sample (Figure S1A). We speculate that the maximum Zr vacancy concentration for achieving stable phase purity is $\sim 12\%$, and $\text{Zr}_{0.88}\text{NiBi}$ is thus designated for further discussion." Figure R1 was also added as Figure S1A in the revised Supplemental Material.

Figure R1. XRD patterns for Zr_{0.88}NiBi and Zr_{0.87}NiBi from different batches synthesized using the same method. Right panel: magnified view between the (111) and (200) peaks.

Comment 2: The discussion about the relation between the twins and the annihilation of impurity phase is obscuring. What are the impurity phases? Why the annihilation of the impurity phases has relation with the twins?

Response: We thank the reviewer for pointing this out. The impurity phases of the unannealed samples are ZrNi alloys based on standard XRD cards (PDF#50-1095 and 38-1170), as shown in Figure R2 (Figure S1B in the revised Supplemental Material). Nonetheless, these impurity phases can be eliminated by employing an additional annealing process that is designed to promote atomic diffusion and reduce the Gibbs free energy of the system. Twin structure, which typically exhibits extremely low interfacial energy, has been demonstrated as the key to the structural stabilization of various face-centered cubic metals (*Nat. Rev. Mater.* 2016, 1, 16019). Although twinned half-Heuslers have rarely been reported, the relevant research on Ni-based alloys could serve as a frame of reference for discussion. Smith *et al.* found that, during the twinning process in Ni-based superalloys, an atomic reordering will occur to eliminate high-energy, wrong nearest-neighbor bonds along the twins (*Nat. Commun.* 2016, 7, 13434). Wu *et al.* revealed an increased frequency of twin boundaries in fully recrystallized Ni-containing high-entropy alloys compared with that of as-cast alloys (*Nat. Commun.* 2022, 13, 4697). We thus speculate that the twinning process and accompanying atomic reordering may contribute to the phase recrystallization in our case.

For clarification, the relevant discussion on page 5 of the revised manuscript has been revised as the following: “Such recrystallization behavior could correlate with thermally activated atomic diffusion, as well as with twinning-process-associated atomic rearrangement^{28,29}, where the formation of twins enables the existence of an interfacial network at a low energy state and thus stabilizes the structure³⁰.” References [28-30] (*Nat. Commun.* 2016, 7, 13434, *Nat. Commun.* 2022, 13, 4697, and *Nat. Rev. Mater.* 2016, 1, 16019, respectively) have been added.

Figure R2. XRD patterns for $Zr_{0.88}NiBi$ and $Zr_{0.90}NiBi$ with and without an additional annealing process.

Comment 3: $Zr_{0.88}NiBi$ and $ZrCoBi$ have similar average atomic mass. Why $ZrCoBi$ has higher sound velocity than $Zr_{0.88}NiBi$?

Response: We thank the reviewer for the comment. By simplifying the lattice as atoms with an average mass of \bar{M} connected to each other by springs (chemical bonds) with a restoring force of F , the sound velocity v_s is determined by $v_s \sim \left(\frac{F}{\bar{M}}\right)^{1/2}$ (*Adv. Mater.* 2018, 30, 1705617). We thus hypothesize weaker chemical bonds in $Zr_{0.88}NiBi$ compared with those in $ZrCoBi$. This hypothesis was validated by our detailed analyses of the differences in the electron localization function, bond length, phonon dispersion, and atomic displacement parameters between $Zr_{0.88}NiBi$ and $ZrCoBi$.

Comment 4: Have the authors calculated the phonon dispersion of $ZrNiBi$ without Zr-vacancy? In principle, $ZrNiBi$ is not stable and imaginary frequency might appear in the phonon dispersion.

Response: We thank the reviewer for pointing this out. Following the reviewer's advice, we further calculated the phonon dispersion of $ZrNiBi$ without Zr vacancy. As shown in Figure R3, no imaginary frequency is observed. It should also be noted that the convex hull distance is another important characteristic for a thermodynamically stable phase. According to the Open Quantum Materials Database (*npj Comput. Mater.* 2015, 1, 15010), the hull distance of $ZrNiBi$ without Zr vacancy is ~ 0.065 eV/atom, indicating that it is not stable. This is also consistent with our experimental XRD results.

Figure R3. Phonon dispersion of ZrNiBi without Zr vacancy.

Comment 5: In Fig. 3, it is better to use the same supercell to calculate the phonon dispersions of $Zr_{0.88}NiBi$ and $ZrCoBi$.

Response: We thank the reviewer for pointing this out. Following the reviewer’s suggestion, we further calculated the phonon dispersion for the supercell of $Zr_8Co_8Bi_8$. For the undefected supercell ($Zr_8Co_8Bi_8$), band crossing can be resolved by estimating band connections from eigenvectors in Phonopy, through which we can obtain the same phonon spectrum as that by using the primitive cell ($ZrCoBi$), as shown in Figure R4. However, for defective $Zr_{0.88}NiBi$, vacancy-induced lattice distortions lead to acoustic-optical hybridized modes, which is a different scenario.

For clarification, we have added Figure R4 as Figure S3 in the revised Supplemental Material and the following has been added on page 7 of the revised manuscript: “In contrast to corresponding results for $ZrCoBi$ (Figure 3F) and phonon dispersion results for the supercell of $Zr_8Co_8Bi_8$ (Figure S3)”.

Figure R4. Phonon dispersions of $Zr_8Co_8Bi_8$ without and with estimations of band connections from eigenvectors in the Phonopy package.

Comment 6: In Fig. 3E, why the Ni atoms have larger ADP than Zr and Bi atoms?

Response: We thank the reviewer for pointing this out. The atomic displacement parameter (ADP) is the mean squared displacement amplitude of the atom relative to its equilibrium position and is used to characterize the strength of the atom’s vibration, which depends on its atomic mass and the frequency of vibration (*Phys. Rev. B* 1997,

55, 10355). In a non-defective cubic half-Heusler phase such as ZrCoBi, the Zr-Co and Co-Bi bonding lengths are identical ($\sim 2.70 \text{ \AA}$), so a lighter constituent atom like Co tends to have a larger ADP. $\text{Zr}_{0.88}\text{NiBi}$, on the other hand, is more complicated. Zr vacancy causes local structural distortion as well as diverse bond lengths ($d_{\text{Ni-Bi}} = 2.67\text{-}2.78 \text{ \AA}$ and $d_{\text{Zr-Ni}} = 2.68\text{-}2.71 \text{ \AA}$). We further plotted the ADPs along the three Cartesian directions for each atom in the supercell of $\text{Zr}_{0.88}\text{NiBi}$ (shown in Figure R5) and found visible fluctuations in the ADPs for each atom (shown in Figure R6). The lighter constituent atom Ni shows the highest averaged ADP at 300 K ($0.73 \times 10^{-2} \text{ \AA}^2$) in comparison to that of Zr ($0.63 \times 10^{-2} \text{ \AA}^2$) and Bi ($0.65 \times 10^{-2} \text{ \AA}^2$). However, such ADP fluctuation is unique and thus deserves greater attention, especially that for Bi, which shows the strongest fluctuation. The smallest ADP for Bi in $\text{Zr}_{0.88}\text{NiBi}$ is broadly comparable to the ADP for Bi in ZrCoBi, indicating the similar vibrating modes in these two materials governed by a global soft bond. Given that the asymmetrically distributed electron cloud appears in the vacancy-neighboring Bi atoms, Bi exhibiting the strongest ADP fluctuation, as well as largest ADP value, is attributed to the local soft bonds that are helpful for understanding the much lower sound velocity of $\text{Zr}_{0.88}\text{NiBi}$ in comparison to that of ZrCoBi.

Figure R5. Atomic displacement parameters (ADPs) along the three Cartesian directions for (A) Zr, (B) Ni, and (C) Bi in the supercell of $\text{Zr}_{0.88}\text{NiBi}$.

Figure R6. Ranges of ADPs for all atoms in the supercell of $Zr_{0.88}NiBi$.

For clarification, the following has been added on page 8 of the revised manuscript: “It should be noted that there are visible fluctuations in the ADPs for each atom in the supercell of $Zr_{0.88}NiBi$, as shown in Figure 3E with details along the three Cartesian directions plotted in Figure S4, which is attributed to the diversity of bond lengths. The ratios of the largest to smallest ADPs for Zr, Ni, and Bi are 1.24:1, 1.12:1, and 1.65:1, respectively, at room temperature. Moreover, the smallest ADP for Bi in $Zr_{0.88}NiBi$ shows only a minor difference compared to the ADP for Bi in $ZrCoBi$, indicating the similar vibrating modes in these two materials governed by a global soft bond. The fact that the largest and second-largest ADPs respectively correspond to Bi5 and Bi4 (both along the y-axis), which both show an asymmetrically distributed electron cloud adjacent to a Zr vacancy (see Figure 3C), attests to the existence of a vacancy-induced much softer local chemical bond.” ADP results in Figure 3E in the revised manuscript were modified as shown in Figure R6, and Figure R5 was added as Figure S4 in the revised Supplemental Material.

Comment 7: In Fig. 4b, the data of $ZrCoBi$ should be also included for comparison.

Response: We appreciate the reviewer’s constructive suggestion. The Pisarenko relation of $ZrCoBi$ has been added in the revised Figure 4B. To further verify the dominant role of transition metals in determining the conductive band edge, the data for $ZrCoSb$ was also included since $ZrCoBi$ and $ZrCoSb$ possess similar DOS effective mass values of $\sim 7 m_e$. In the revised manuscript, the following was added on page 9: “Additionally, $ZrCoSb$ and $ZrCoBi$ also display identical behavior^{50,51}, further confirming the negligible influence of the main group element on the CBM.” Figure 4B was modified as shown in Figure R7.

Figure R7. Pisarenko relation of $\text{Zr}_{0.88}\text{NiBi}$ in comparison to that of ZrNiSn , ZrNiPb , ZrCoSb , and ZrCoBi . Symbols: experimental data. Dashed lines: obtained from the SPB model.

Comment 8: I suggest the authors to comment how to further optimize the TE performance of defective half-Heusler alloys.

Response: We thank the reviewer for the constructive suggestion. As shown in Figure 1, the lattice thermal conductivity (κ_{lat}) of most defective half-Heuslers remains higher than that of other outstanding thermoelectric materials, encouraging further κ_{lat} reduction. Given that defective ZrNiBi exhibits the lowest κ_{lat} among the family of half-Heuslers, we modeled its thermoelectric performance as plotted in Figure R8 for discussion. It is clear that additional phonon and electron concentration engineering will contribute to performance enhancement. More intriguingly, vacancies in this material are randomly distributed, while vacancies in some other defective half-Heuslers have short-range order (*Energy Environ. Sci.* 2019, 12, 1568). We found that vacancy-ordering manipulation enables higher performance.

Following the reviewer's suggestion, the following discussion has been added on pages 11-12 of the revised manuscript: "Given that defects have been regarded as an additional degree of freedom for boosting thermoelectric performance, studies of the vacancy-abundant half-Heuslers have the potential to reshape the development of high-temperature thermoelectric materials. Here we modeled the figure of merit of the defective ZrNiBi at 973 K. When the electron transport of defective ZrNiBi reaches the Ioffe–Regel limit (electron mean free path $\sim a$, where a denotes the lattice constant)⁵⁴ and its phonon transport follows the Cahill model (minimal lattice thermal conductivity $\sim 0.66 \text{ W m}^{-1} \text{ K}^{-1}$)⁶⁰, its figure of merit exhibits an impressive value of ~ 2.2 at the electron concentration of $4\text{--}5 \times 10^{20} \text{ cm}^{-3}$ (Figure S7A). It should be noted that our transmission electron microscope results revealed a random distribution of vacancies in $\text{Zr}_{0.88}\text{NiBi}$, in contrast to some typical defective 19-electron half-Heuslers that have

been reported to show short-range order among their vacancies (*e.g.*, $\text{Nb}_{0.8}\text{CoSb}^{22}$). It is thus natural to examine whether the performance of defective ZrNiBi could be enhanced if its degree of vacancy order can be improved. However, the existence of vacancy pairs and vacancy triplets complicates the short-range order in defective 19-electron half-Heusler systems²². Since long-range order is an extreme case of vacancy order, we assumed the existence of long-range-ordered vacancies in $\text{Zr}_{0.88}\text{NiBi}$ (where the shortest inter-vacancy distance of $\sqrt{2}a$ is assigned to the electron mean free path) and, correspondingly, the material's figure of merit as a function of both lattice thermal conductivity and electron concentration was plotted in Figure S7B. In comparison to the framework of the Ioffe–Regel condition, the optimal electron concentration is shifted to a lower range and the material exhibits better performance, *e.g.*, its peak figure of merit is enhanced to ~ 2.7 as the lattice thermal conductivity decreases to the value estimated by the Cahill model. It must be noted that a system with only long-range-ordered vacancies is hardly achievable based on experimental results, so the trend found from our model sets an upper bound for the thermoelectric performance of defective ZrNiBi while the degree of vacancy order can serve as a control for its figure of merit. In fact, current optimization approaches have resulted in a peak figure of merit of ~ 1.0 for a $\text{Zr}_{0.88}\text{NiBi}$ -based material at a lattice thermal conductivity of $\sim 0.9 \text{ W m}^{-1} \text{ K}^{-1}$ and an electron concentration of $\sim 1.5 \times 10^{21} \text{ cm}^{-3}$. To fulfill its thermoelectric potential, delicate phonon and electron concentration engineering (toward lower values for both lattice thermal conductivity and electron concentration) is necessary. Additionally, vacancy manipulation for ordered distribution enables further enhancement of performance as depicted in Figure S7C.” Figure R8 was added as Figure S7 in the revised Supplemental Material.

Figure R8. Modeled figure of merit for defective ZrNiBi at 973 K. Dependence of figure of merit on lattice thermal conductivity and electron concentration based on (A) the Ioffe–Regel condition and the Cahill model and (B) the long-range ordering condition and the Cahill model. (C) Comparison between experimentally determined performance (symbols) and modeled performance (solid lines). Gray bar: electron concentration range for figure of merit of ~ 2.2 determined by the Ioffe–Regel limit and the Cahill model.

Response to Reviewer 2:

Reviewer #2: The authors studied how vacancy affected the thermoelectric properties of defective half-Hausler ZrNiBi. I appreciate the detailed study the authors carried out by combining experiments and computation. It would be an important piece of work for the community. I have the following questions/comments for the authors to consider:

Response: We greatly appreciate the reviewer’s time and effort in reading our manuscript, understanding its value, and providing valuable suggestions. Following are our point-by-point responses to the reviewer’s constructive comments.

Comment 1: The biggest question I have is what really is “anomalous.” I feel all the results were expected. The theoretical treatment of vacancy has been challenging, so people often used the empirical equation to treat it as a phonon scattering center with six times the mass of the original atom. But it doesn’t mean that people are unaware of the effects of softening bonds, lowering group velocities, etc. For example, using the virtual crystal model for alloys can lead to overestimating the phonon lifetimes if the local force constants variations were not included [Journal of Physics: Condensed Matter 27 (37), 375403]. In this sense, I would appreciate it if the authors could better cultivate/support their claim of “anomalous.” Or maybe it is better to remove “anomalous” and claim a comprehensive understanding of

Response: We thank the reviewer for the comment. It is true that both intrinsic and extrinsic defects will cause perturbation of the force constants. However, vacancy-induced bond softening is not the whole story in cases where an opposite effect (*i.e.*, bond strengthening) has also been reported. Based on Wuttig *et al.*’s study of GeSbTe-based alloys (*Nat. Mater.* 2007, 6, 122), both Ge-Te and Sb-Te bonds are strengthened because the formation of vacancies reduces the antibonding characteristics. Lee *et al.*’s charge density and electron localization function results show that vacancy-containing GeTe exhibits a reduced proportion of weak hyperbonds in comparison to its vacancy-free counterpart (*Adv. Mater.* 2020, 32, 2000340). We think the correlation between vacancies (as well as other defects) and material properties is complicated and highly material-dependent, making it difficult to determine, especially for a new material with abundant unknown properties.

The emerging defective ZrNiBi is distinctive among the family of half-Heuslers since its phononic and electronic transport properties cannot be fully explained using the common transport pictures for describing half-Heuslers, and they are highly correlated

to the intrinsic vacancies. Regarding phononic transport, defective ZrNiBi possesses the lowest lattice thermal conductivity among the half-Heuslers, and this is not solely reliant on the defect scattering. Its anomalously low sound velocity, as depicted in Figure 3B, also plays a role. However, a heavy average atomic mass and a global soft bond fail to elucidate the origin of such low sound velocity. Our detailed study of the phonon property of defective ZrNiBi not only unraveled the hidden role of local soft bonding in suppressing its sound velocity, but also revealed various unique features that have not been explored in the previously reported half-Heuslers, such as bond length diversity, anisotropic atomic displacement, and strongly hybridized acoustic-optical modes.

Regarding electronic transport, vacancy-induced band flattening features prominently in the conductive band of defective ZrNiBi. As a result, it has an increased density of states effective mass ($\sim 6.0 m_e$) in comparison with that of ZrNiSn and ZrNiPb ($\sim 3.0 m_e$), which is also counterintuitive since the conductive band minimum of a half-Heusler is primarily governed by the interaction between the d -orbitals of two transition metals. This further signifies that vacancy can serve as another degree of freedom for electronic band engineering in discovering novel transport behaviors and exploring diverse electronic applications.

Comment 2: In the “Theoretical calculation” section, “A $2 \times 2 \times 2$ supercell with 23 atoms following removal of one Zr atom was used to calculate the electronic structure of $Zr_{0.875}NiBi$.” Were 23 atoms the number of atoms in the unit cell? Please clarify.

Response: We thank the reviewer for pointing this out. The primitive cell of ZrNiBi contains one Zr atom, one Ni atom, and one Bi atom. A $2 \times 2 \times 2$ supercell has 24 atoms ($Zr_8Ni_8Bi_8$). The electronic structure of $Zr_{0.875}NiBi$ was calculated by further removing one Zr atom, leaving 23 atoms ($Zr_7Ni_8Bi_8$).

For clarification, the following has been updated in the “Theoretical calculation” section on page 24 of the revised manuscript: “A $2 \times 2 \times 2$ supercell with 23 atoms following removal of one Zr atom (*i.e.*, $Zr_7Ni_8Bi_8$) was used to calculate the electronic structure of $Zr_{0.875}NiBi$.”

Comment 3: Can the authors comment on how much higher they expect ZT to reach if vacancies are optimized to promote the thermoelectric properties?

Response: We appreciate the reviewer’s constructive suggestion. We first modeled the figure of merit of defective ZrNiBi based on the Ioffe–Regel condition and the Cahill model, and found that peak ZT can reach ~ 2.2 at 973 K. In this case, vacancies are randomly distributed, consistent with our experimental observation. When the vacancies are optimized toward long-range-ordered distribution, the electron mean free path is assumed to be the shortest distance between the vacancies (*i.e.*, $\sqrt{2}a$, where a

denotes the lattice constant) and peak ZT can be further enhanced to ~ 2.7 at 973 K. It must be noted that delicate phonon and electron concentration engineering is essential to optimize performance regardless of which transport model is applied.

For clarification, the modeled ZT shown in Figure R8 has been added as Figure S7 in the revised Supplemental Material and the following discussion has also been added on pages 11-12 of the revised manuscript: “Given that defects have been regarded as an additional degree of freedom for boosting thermoelectric performance, studies of the vacancy-abundant half-Heuslers have the potential to reshape the development of high-temperature thermoelectric materials. Here we modeled the figure of merit of the defective ZrNiBi at 973 K. When the electron transport of defective ZrNiBi reaches the Ioffe–Regel limit (electron mean free path $\sim a$, where a denotes the lattice constant)⁵⁴ and its phonon transport follows the Cahill model (minimal lattice thermal conductivity $\sim 0.66 \text{ W m}^{-1} \text{ K}^{-1}$)⁶⁰, its figure of merit exhibits an impressive value of ~ 2.2 at the electron concentration of $4\text{--}5 \times 10^{20} \text{ cm}^{-3}$ (Figure S7A). It should be noted that our transmission electron microscope results revealed a random distribution of vacancies in $\text{Zr}_{0.88}\text{NiBi}$, in contrast to some typical defective 19-electron half-Heuslers that have been reported to show short-range order among their vacancies (*e.g.*, $\text{Nb}_{0.8}\text{CoSb}$ ²²). It is thus natural to examine whether the performance of defective ZrNiBi could be enhanced if its degree of vacancy order can be improved. However, the existence of vacancy pairs and vacancy triplets complicates the short-range order in defective 19-electron half-Heusler systems²². Since long-range order is an extreme case of vacancy order, we assumed the existence of long-range-ordered vacancies in $\text{Zr}_{0.88}\text{NiBi}$ (where the shortest inter-vacancy distance of $\sqrt{2}a$ is assigned to the electron mean free path) and, correspondingly, the material’s figure of merit as a function of both lattice thermal conductivity and electron concentration was plotted in Figure S7B. In comparison to the framework of the Ioffe–Regel condition, the optimal electron concentration is shifted to a lower range and the material exhibits better performance, *e.g.*, its peak figure of merit is enhanced to ~ 2.7 as the lattice thermal conductivity decreases to the value estimated by the Cahill model. It must be noted that a system with only long-range-ordered vacancies is hardly achievable based on experimental results, so the trend found from our model sets an upper bound for the thermoelectric performance of defective ZrNiBi while the degree of vacancy order can serve as a control for its figure of merit. In fact, current optimization approaches have resulted in a peak figure of merit of ~ 1.0 for a $\text{Zr}_{0.88}\text{NiBi}$ -based material at a lattice thermal conductivity of $\sim 0.9 \text{ W m}^{-1} \text{ K}^{-1}$ and an electron concentration of $\sim 1.5 \times 10^{21} \text{ cm}^{-3}$. To fulfill its thermoelectric potential, delicate phonon and electron concentration engineering (toward lower values for both lattice thermal conductivity and electron concentration) is necessary. Additionally, vacancy manipulation for ordered distribution enables further enhancement of performance as depicted in Figure S7C.”

Comment 4: I wasn’t sure why the discussion about “twinning” was relevant to the

vacancy.

Response: We thank the reviewer for the comment. Based on Xia *et al.*'s study (*Energy Environ. Sci.* 2019, 12, 1568), short-range-ordered vacancies were observed in a series of defective 19-electron half-Heuslers, but no twin was observed. They claimed that the short-range-ordered vacancy structure is at a lower energy state. In contrast, defective ZrNiBi studied here lacks this short-range ordered structure but features a twinning structure that can reduce the total energy of the interfacial network. There might be a competing mechanism between these two atomic configurations in terms of the structure stabilization energy. In addition, Zhang *et al.* reported a positive effect of short-range order (SRO) on stacking fault energy (*Nature* 2020, 581, 283) while twins are more easily formed in face-centered-cubic materials with low stacking fault energy.

For clarification, the following has been updated at the end of the “Crystallographic features” section on pages 5-6 of the revised manuscript: “Xia *et al.* found that the SRO configuration favors a lower energy state²², but the twinning configuration shows similar efficacy. Additionally, stacking fault energy could be another coupling factor since SRO has a positive influence on it³¹, while twins are more easily formed in face-centered-cubic materials with low stacking fault energy³⁰. As a result, competition between these two atomic configurations may occur. It should also be noted that the microstructure is highly sensitive to the synthesis process used, which allows for the possibility to finely control the atomic configuration, as well as the structure–property relationship, of defective HHs.” Reference [31] (*Nature* 2020, 581, 283) has been added.

Comment 5: The phonon dispersion features of no acoustic-optical bandgap are quite similar to hybrid perovskites that possess ultralow thermal conductivity. It would be nice to include or comment on that. [Physical review letters 123 (15), 155901; Nano Letters 21 (9), 3708-3714].

Response: We thank the reviewer for the constructive suggestion. Both reports mentioned by the reviewer demonstrate that the frequency overlap between acoustic and optical phonons enables strong acoustic-optical scattering as well as low lattice thermal conductivity. This result is a useful supplement to our discussion about the contributing role of optical-phonon-involved high-order phonon scattering in heat transport.

Following the reviewer's suggestion, we updated the following statement on page 8 of the revised manuscript: “The prevailing viewpoint is that thermal resistance results from the lowest-order three-phonon scattering, which gives rise to a T^{-1} dependence of κ_{lat} ³⁴. However, a higher-order phonon scattering process considering optical phonons has gained increasing attention and its adequacy has also been proved in other materials such as boron arsenide⁴²⁻⁴⁴ and hybrid perovskites^{45,46}.” References [45,46] (*Phys. Rev. Lett.* 2019, 123, 155901, and *Nano Lett.* 2021, 21, 3708, respectively) have been added.

Response to Reviewer 3:

Reviewer #3: The present manuscript reported on a defective ZrNiBi HH alloy with an ultralow lattice thermal conductivity and a high ZT value. The low κ lattice was attributed to the soft chemical bonds, acoustic-optical phonon coupling, and point defect scattering, etc. induced by the presence of Zr vacancies. A vacancy-induced flat band was also observed, which leads to much-improved density-of-states effective mass and power factor. By introducing proper dopants to tune the carrier concentration, a peak ZT of 1.0 at 973 K is obtained. The conclusions are supported by comprehensive structural characterizations and theoretical calculations. This work reveals the multifunctional properties of vacancies in mediating anomalous phononic and electronic transports. It could provide a new paradigm for property modifications by controlling vacancies. Thus, I recommend the publication of this work in Nature Communications with minor revisions. My specific comments are listed below.

Response: We greatly appreciate the reviewer's time and effort in reading our manuscript, understanding its value, and providing valuable suggestions. Following are our point-by-point responses to the reviewer's constructive comments.

Comment 1: The authors emphasized the observation of twinning structures but did not discuss the impact of twinning on the thermoelectric properties. It has been reported in many studies that the twin structure can influence the electronic band structure and phonon scattering.

Response: We thank the reviewer for pointing this out. According to previous research (*Nano Energy* 2015, 17, 279, *Nano Energy* 2017, 37, 203), a major way twinning structure influences transport properties is the carrier filtering effect, where most carriers with energy above the barrier height can pass through the twin boundary while others with energy lower than the barrier height are scattered. This effect will contribute to Seebeck coefficient enhancement and hardly affects carrier mobility. In the material studied here, the electron mean free path is comparable to the lattice constant, indicating that electrons are scattered before the filtering effect occurs. We thus speculate that the carrier filtering effect could be negligible. For clarification, the following has been added on page 10 of the revised manuscript: "This also implies that the carrier filtering effect from twin boundaries could be negligible since electrons are scattered before this effect occurs."

With respect to phonon transport, the scattering effect of a twinning structure remains a subject of debate. The results of some calculations and experiments have indicated that the contribution of twin boundaries to phonon scattering is significant, while other reports have argued that they have a weaker scattering effect than grain boundaries (*Adv. Phys.* 2018, 67, 69). It should be noted that most twinned crystals have been reported with coherent twin boundaries, and such coherence is suspected to induce minor thermal boundary resistance. In $Zr_{0.88}NiBi$, the formation of the secondary nanotwin

originates from the defect (kink) on the twin boundary, implying that the compound's lattice coherence has been interrupted. It is thus reasonable that the twofold twinning configuration will contribute to effective phonon scattering. This was discussed in the first paragraph of the "Phononic dynamics" section in the original manuscript.

Therefore, we conclude that the impact of twinning structure on the thermoelectric performance of $\text{Zr}_{0.88}\text{NiBi}$ mainly arises from its contribution to the material's low lattice thermal conductivity given its minor effect on the material's electrical property.

Comment 2: The local soft chemical bonding was derived from the electron localization from ELF values. I do not understand the correlations between a large ELF and a soft chemical bond. This seems to contradict the intuition that strong electron localization is generally obtained in stiff bonds such as covalent and ionic bonds. In contrast, the soft metallic and metavalent bonds often show a strong electron delocalization.

Response: We thank the reviewer for the comment. The reviewer is correct that strong electron localization corresponds to stiff bonds like covalent and ionic bonds. In their study of RhBi_4 (*Angew. Chem. Int. Ed.* 1995, 34, 1204), Grin *et al.* found that the Bi atoms exhibit high ELF values as well and concluded that this originates from the material's lone pairs of Bi atoms. Meanwhile, the mechanical property results for ZrCoBi reveal its global soft bonds, for which the strong relativistic effect of Bi contracts the 6s shell and increases its inertness for bonding (*Nat. Commun.* 2018, 9, 2497). We thus consider that $\text{Zr}_{0.88}\text{NiBi}$ has a similar characteristic. Furthermore, we found diverse bond lengths in this highly symmetrical material ($d_{\text{Ni-Bi}} = 2.67\text{-}2.78 \text{ \AA}$ and $d_{\text{Zr-Ni}} = 2.68\text{-}2.71 \text{ \AA}$, as shown in Figure R9), which enables large and anisotropic atomic displacement (shown in Figure R5 and R6). These findings reinforce our viewpoint of its soft bonding nature.

To avoid misleading readers, we updated the discussion of ELF and added bond length and atomic displacement parameter analyses to support our viewpoint as detailed below. Figure 3C and D were modified as shown in Figure R9.

The following has been updated on pages 6-7 of the revised manuscript: "Additionally, the relativistic contraction of the Bi-6s orbital enables inert chemical bonds³⁶ (*i.e.*, a global weak interatomic force). It should also be noted that, regardless of their similar \bar{M} or the nature of Bi, $\text{Zr}_{0.88}\text{NiBi}$ is distinguished among the heavy-element-containing HHs by its anomalously low v_s . We assumed a much weaker interatomic force in $\text{Zr}_{0.88}\text{NiBi}$ and then calculated its electron localization function (ELF) and bond lengths to investigate its bonding characteristics ($2 \times 2 \times 2$ supercell with removal of one Zr atom corresponding to $\text{Zr}_{0.875}\text{NiBi}$, hereafter approximated as $\text{Zr}_{0.88}\text{NiBi}$). The ELF value ranges from 0 to 1, where $\text{ELF} = 1$ indicates complete localization of electrons. Figure 3C and D illustrate the projected two-dimensional ELFs along the $(01\bar{1})$ plane for $\text{Zr}_{0.88}\text{NiBi}$ and ZrCoBi , respectively, in both of which thin grey circle-like contour lines

denote regions with the ELF value of 0.8, thus revealing strong electron localization in the Bi atoms for both materials. This phenomenon is similar to that observed in RhBi_4 , which was attributed to the material's lone pairs of Bi atoms⁴⁰. More intriguingly, asymmetric distribution of the electron cloud in the Bi atoms that adjoin Zr vacancies is observed in $\text{Zr}_{0.88}\text{NiBi}$, as evidenced by the existence of regions with the higher ELF value of 0.85 (denoted by the thick black crescent-like contour lines in Figure 3C) in the direction of the vacancies. As a result, a local structural distortion occurs and is accompanied by diverse bond lengths ($d_{\text{Ni-Bi}} = 2.67\text{-}2.78 \text{ \AA}$ and $d_{\text{Zr-Ni}} = 2.68\text{-}2.71 \text{ \AA}$), in contrast to ZrCoBi , which shows identical Co-Bi and Zr-Co bond lengths ($\sim 2.70 \text{ \AA}$). Such diversity in bond lengths permits large and anisotropic atomic displacement, which is a typical indicator of soft bonds. Atomic displacement parameters (ADPs) will be discussed later. Both the appearance of asymmetric electron localization in the vacancy-neighboring Bi atoms in $\text{Zr}_{0.88}\text{NiBi}$ and the slight changes in its Zr-Ni bond lengths indicate the local softening of chemical bonds in this case, and this can also be ascertained from the results of ADP analysis discussed below.”

The following has been added on page 8 of the revised manuscript: “It should be noted that there are visible fluctuations in the ADPs for each atom in the supercell of $\text{Zr}_{0.88}\text{NiBi}$, as shown in Figure 3E with details along the three Cartesian directions plotted in Figure S4, which is attributed to the diversity of bond lengths. The ratios of the largest to smallest ADPs for Zr, Ni, and Bi are 1.24:1, 1.12:1, and 1.65:1, respectively, at room temperature. Moreover, the smallest ADP for Bi in $\text{Zr}_{0.88}\text{NiBi}$ shows only a minor difference compared to the ADP for Bi in ZrCoBi , indicating the similar vibrating modes in these two materials governed by a global soft bond. The fact that the largest and second-largest ADPs respectively correspond to Bi5 and Bi4 (both along the y-axis), which both show an asymmetrically distributed electron cloud adjacent to a Zr vacancy (see Figure 3C), attests to the existence of a vacancy-induced much softer local chemical bond.”

Figure R9. Calculated two-dimensional ELF maps along the $(01\bar{1})$ plane together with corresponding bond lengths for $\text{Zr}_{0.88}\text{NiBi}$ (left) and ZrCoBi (right). Thin grey circle-like and thick black crescent-like contour lines correspond to the ELF values of 0.8 and 0.85, respectively.

Comment 3: The authors described the coupling of low-lying optical phonons and acoustic phonons in sample $\text{Zr}_{0.88}\text{NiBi}$. Yet, it is not very convenient to observe this phenomenon in the calculated phonon spectrum. It would be helpful if the corresponding phonon branches can be indicated by different colors or by arrows. I do

not see obvious TO softening at the Γ point compared with the typical phenomenon observed in anharmonic PbTe.

Response: We thank the reviewer for the constructive suggestion. 1) Vacancy causes lattice distortion in $\text{Zr}_{0.88}\text{NiBi}$, which leads to strong acoustic-optic hybridized modes in the frequency range of 1.87-2.22 THz. As an example, based on eigenvectors, the L point is two degenerate acoustic modes around 1.37 THz and then becomes an optical mode around 1.63 THz. However, at 1.87, 2.10, 2.16, and 2.22 THz, the acoustic modes strongly hybridize with the optical ones. Therefore, it is difficult to separate the acoustic and optical branches away from the Γ point in acoustic-optic hybridized spectra. 2) Half-Heusler compounds are well known for their good electrical properties but with intrinsically high lattice thermal conductivity. Therefore, unlike for PbTe, it is rare for half-Heusler compounds to exhibit obvious TO softening at the Γ point (*Nat. Mater.* 2011, 10, 614). However, low-lying optical modes depress the acoustic branches, which is an important feature for intrinsically low lattice thermal conductivity, such as in LiZnSb ($\kappa_{\text{lat}} = 1.79 \text{ W m}^{-1} \text{ K}^{-1}$ at 300 K) (*J. Phys. Chem. C* 2019, 123, 18824). Figure R10 shows the phonon dispersion of LiZnSb in comparison to that of NbFeSb with intrinsically high lattice thermal conductivity ($\kappa_{\text{lat}} = 22 \text{ W m}^{-1} \text{ K}^{-1}$ at 300 K) (*Phys. Chem. Chem. Phys.* 2017, 19, 4411).

For clarification, the following statement has been updated on page 8 of the revised manuscript: “ $\text{Zr}_{0.88}\text{NiBi}$ features low-lying optical phonons (1.5–3 THz) that are strongly hybridized with acoustic branches (in the frequency range between 1.87 and 2.22 THz).”

Figure R10. Phonon dispersions of NbFeSb and LiZnSb.

List of Changes

	Action	Location	Content
1	Addition	Page 4, revised manuscript. Figure S1A, Page 5, revised Supplemental Material.	Added: “It should be noted that the XRD patterns for $Zr_{0.87}NiBi$ samples from two different batches showed contrasting phase purity despite the identical synthesis method used for each, where the sample from Batch I was found to be phase-pure while a tiny amount of Bi impurity was observed in the Batch II sample (Figure S1A). We speculate that the maximum Zr vacancy concentration for achieving stable phase purity is ~12%, and $Zr_{0.88}NiBi$ is thus designated for further discussion.” Added Figure S1A.
2	Change; reference addition and renumbering	Page 5, revised manuscript.	Original: “Such recrystallization behavior could correlate with the formation of twins.” Revised: “Such recrystallization behavior could correlate with thermally activated atomic diffusion, as well as with twinning-process-associated atomic rearrangement ^{28,29} , where the formation of twins enables the existence of an interfacial network at a low energy state and thus stabilizes the structure ³⁰ .” Added references [28-30] and renumbered subsequent references accordingly.
3	Change; reference addition and renumbering	End of “Crystallographic features” section, Pages 5-6, revised manuscript.	Abbreviated “short-range order” to “SRO”. Original: “The breaking of short-range order could be attributed to the twinning process. Therefore, the extraordinary structural properties of $Zr_{0.88}NiBi$ make

			this material crystallographically attractive.” Revised: “Xia et al. found that the SRO configuration favors a lower energy state²², but the twinning configuration shows similar efficacy. Additionally, stacking fault energy could be another coupling factor since SRO has a positive influence on it³¹, while twins are more easily formed in face-centered-cubic materials with low stacking fault energy³⁰. As a result, competition between these two atomic configurations may occur. It should also be noted that the microstructure is highly sensitive to the synthesis process used, which allows for the possibility to finely control the atomic configuration, as well as the structure–property relationship, of defective HHs.” Added reference [31] and renumbered subsequent references accordingly.
4	Change; reference addition and renumbering	Pages 6-7, revised manuscript. Figure 3C and D, Page 16, revised manuscript.	Updated discussion of electron localization function as follows: “Additionally, the relativistic contraction of the Bi-6s orbital enables inert chemical bonds³⁶ (i.e., a global weak interatomic force). It should also be noted that, regardless of their similar \bar{M} or the nature of Bi, $Zr_{0.88}NiBi$ is distinguished among the heavy-element-containing HHs by its anomalously low v_s. We assumed a much weaker interatomic force in $Zr_{0.88}NiBi$ and then calculated its electron localization function (ELF) and bond lengths to investigate its bonding characteristics ($2 \times 2 \times 2$ supercell with

			removal of one Zr atom corresponding to $Zr_{0.875}NiBi$, hereafter approximated as $Zr_{0.88}NiBi$). The ELF value ranges from 0 to 1, where $ELF = 1$ indicates complete localization of electrons. Figure 3C and D illustrate the projected two-dimensional ELFs along the $(01\bar{1})$ plane for $Zr_{0.88}NiBi$ and $ZrCoBi$, respectively, in both of which thin grey circle-like contour lines denote regions with the ELF value of 0.8, thus revealing strong electron localization in the Bi atoms for both materials. This phenomenon is similar to that observed in $RhBi_4$, which was attributed to the material's lone pairs of Bi atoms⁴⁰. More intriguingly, asymmetric distribution of the electron cloud in the Bi atoms that adjoin Zr vacancies is observed in $Zr_{0.88}NiBi$, as evidenced by the existence of regions with the higher ELF value of 0.85 (denoted by the thick black crescent-like contour lines in Figure 3C) in the direction of the vacancies. As a result, a local structural distortion occurs and is accompanied by diverse bond lengths ($d_{Ni-Bi} = 2.67-2.78$ Å and $d_{Zr-Ni} = 2.68-2.71$ Å), in contrast to $ZrCoBi$, which shows identical Co-Bi and Zr-Co bond lengths (~ 2.70 Å). Such diversity in bond lengths permits large and anisotropic atomic displacement, which is a typical indicator of soft bonds. Atomic displacement parameters (ADPs) will be discussed later. Both the appearance of asymmetric electron localization in the vacancy-neighboring Bi atoms in $Zr_{0.88}NiBi$ and the slight
--	--	--	---

			changes in its Zr-Ni bond lengths indicate the local softening of chemical bonds in this case, and this can also be ascertained from the results of ADP analysis discussed below.” Updated Figure 3C and D to include bond lengths. Added reference [40] and renumbered subsequent references accordingly.
5	Addition	Page 7, revised manuscript. Figure S3, Page 7, revised Supplemental Material.	Added: “In contrast to corresponding results for ZrCoBi (Figure 3F) and phonon dispersion results for the supercell of Zr₈Co₈Bi₈ (Figure S3)”. Added Figure S3.
6	Change	Page 8, revised manuscript.	Original: “Zr_{0.88}NiBi features low-lying optical phonons (1.5–3 THz) that are strongly coupled with acoustic branches.” Revised: “Zr_{0.88}NiBi features low-lying optical phonons (1.5–3 THz) that are strongly hybridized with acoustic branches (in the frequency range between 1.87 and 2.22 THz).”
7	Change; reference addition and renumbering	Page 8, revised manuscript.	Original: “The prevailing viewpoint is that thermal resistance results from the inherent three-phonon Umklapp scattering, which gives rise to a T^{-1} dependence of κ_{lat}. However, a four-phonon scattering process considering optical phonons has gained increasing attention and its adequacy has also been proved in other materials such as boron arsenide³⁷⁻³⁹.” Revised: “The prevailing viewpoint is that thermal resistance results from the lowest-order three-phonon scattering, which gives rise to a T^{-1} dependence of κ_{lat}³⁴. However, a higher-order phonon

			scattering process considering optical phonons has gained increasing attention and its adequacy has also been proved in other materials such as boron arsenide⁴²⁻⁴⁴ and hybrid perovskites^{45,46}.” Added references [45,46] and renumbered subsequent references accordingly.
8	Addition and change	Page 8, revised manuscript. Figure 3E, Page 16, revised manuscript. Figure S4, Page 8, revised Supplemental Material.	Added: “It should be noted that there are visible fluctuations in the ADPs for each atom in the supercell of $Zr_{0.88}NiBi$, as shown in Figure 3E with details along the three Cartesian directions plotted in Figure S4, which is attributed to the diversity of bond lengths. The ratios of the largest to smallest ADPs for Zr, Ni, and Bi are 1.24:1, 1.12:1, and 1.65:1, respectively, at room temperature. Moreover, the smallest ADP for Bi in $Zr_{0.88}NiBi$ shows only a minor difference compared to the ADP for Bi in $ZrCoBi$, indicating the similar vibrating modes in these two materials governed by a global soft bond. The fact that the largest and second-largest ADPs respectively correspond to Bi5 and Bi4 (both along the y-axis), which both show an asymmetrically distributed electron cloud adjacent to a Zr vacancy (see Figure 3C), attests to the existence of a vacancy-induced much softer local chemical bond.” Revised Figure 3E to show ranges of ADPs for all atoms in the supercell of $Zr_{0.88}NiBi$. Added Figure S4.
9	Addition and change	Page 9, revised manuscript.	Added: “Additionally, $ZrCoSb$ and $ZrCoBi$ also display identical

		Figure 4B, Page 17, revised manuscript.	behavior ^{50,51} , further confirming the negligible influence of the main group element on the CBM.” Revised Figure 4B to include Pisarenko relations of ZrCoSb and ZrCoBi.
10	Addition	Page 10, revised manuscript.	Added: “This also implies that the carrier filtering effect from twin boundaries could be negligible since electrons are scattered before this effect occurs.”
11	Addition	Page 11, revised manuscript.	Added: “additionally, the phase purity of all $Zr_{0.88}Ni_{1-x}Co_xBi_{1-y}Sb_y$ samples studied is shown in Figure S1C”.
12	Addition; reference addition and renumbering	Pages 11-12, revised manuscript. Figure S7, Page 11, revised Supplemental Material.	Added: “Given that defects have been regarded as an additional degree of freedom for boosting thermoelectric performance, studies of the vacancy-abundant half-Heuslers have the potential to reshape the development of high-temperature thermoelectric materials. Here we modeled the figure of merit of the defective ZrNiBi at 973 K. When the electron transport of defective ZrNiBi reaches the Ioffe–Regel limit (electron mean free path $\sim a$, where a denotes the lattice constant) ⁵⁴ and its phonon transport follows the Cahill model (minimal lattice thermal conductivity $\sim 0.66 \text{ W m}^{-1} \text{ K}^{-1}$) ⁶⁰ , its figure of merit exhibits an impressive value of ~ 2.2 at the electron concentration of $4\text{--}5 \times 10^{20} \text{ cm}^{-3}$ (Figure S7A). It should be noted that our transmission electron microscope results revealed a random distribution of vacancies in $Zr_{0.88}NiBi$, in contrast to some typical defective 19-electron half-Heuslers that have been reported to

			show short-range order among their vacancies (e.g., Nb_{0.8}CoSb²²). It is thus natural to examine whether the performance of defective ZrNiBi could be enhanced if its degree of vacancy order can be improved. However, the existence of vacancy pairs and vacancy triplets complicates the short-range order in defective 19-electron half-Heusler systems²². Since long-range order is an extreme case of vacancy order, we assumed the existence of long-range-ordered vacancies in Zr_{0.88}NiBi (where the shortest inter-vacancy distance of $\sqrt{2}a$ is assigned to the electron mean free path) and, correspondingly, the material's figure of merit as a function of both lattice thermal conductivity and electron concentration was plotted in Figure S7B. In comparison to the framework of the Ioffe–Regel condition, the optimal electron concentration is shifted to a lower range and the material exhibits better performance, e.g., its peak figure of merit is enhanced to ~ 2.7 as the lattice thermal conductivity decreases to the value estimated by the Cahill model. It must be noted that a system with only long-range-ordered vacancies is hardly achievable based on experimental results, so the trend found from our model sets an upper bound for the thermoelectric performance of defective ZrNiBi while the degree of vacancy order can serve as a control for its figure of merit. In fact, current optimization approaches have resulted in a peak
--	--	--	--

			figure of merit of ~ 1.0 for a $Zr_{0.88}NiBi$-based material at a lattice thermal conductivity of $\sim 0.9 \text{ W m}^{-1} \text{ K}^{-1}$ and an electron concentration of $\sim 1.5 \times 10^{21} \text{ cm}^{-3}$. To fulfill its thermoelectric potential, delicate phonon and electron concentration engineering (toward lower values for both lattice thermal conductivity and electron concentration) is necessary. Additionally, vacancy manipulation for ordered distribution enables further enhancement of performance as depicted in Figure S7C.” Added reference [60] and renumbered subsequent references accordingly. Added Figure S7.
13	Addition	“Theoretical calculation” section. Page 24, revised manuscript.	Added: “... (i.e., $Zr_7Ni_8Bi_8$) ...”

REVIEWERS' COMMENTS

Reviewer #1 (Remarks to the Author):

The authors have satisfactorily answered all the comments of the reviewer. I recommend it for publication without further review.

Reviewer #2 (Remarks to the Author):

The authors have satisfactorily addressed my comments.

Reviewer #3 (Remarks to the Author):

The authors have addressed my comments properly. The manuscript can be published now. Congrats on this nice work.

Reviewer #1:

The authors have satisfactorily answered all the comments of the reviewer. I recommend it for publication without further review.

Response: We appreciate the reviewer's recommendation.

Reviewer #2:

The authors have satisfactorily addressed my comments.

Response: We appreciate the reviewer's positive comments.

Reviewer #3:

The authors have addressed my comments properly. The manuscript can be published now. Congrats on this nice work.

Response: We appreciate the reviewer's recommendation.